# Validation of the German Emotional Contagion Scale and development of a mimicry brief version

Tobias Janelt[1]*, Tobias Altmann[1], Danièle Anne Gubler[2], Marcus Roth[1]

**1** Department of Psychology, University of Duisburg-Essen, Essen, Germany, **2** Department of Psychology, University of Bern, Bern, Switzerland

* tobias.janelt@uni-due.de

## Abstract

The susceptibility to emotional contagion has been psychometrically addressed by the self-reported Emotional Contagion Scale. With the present research, we validated a German adaptation of this scale and developed a mimicry brief version by selecting only the four items explicitly addressing the overt subprocess of mimicry. Across three studies ($N_1 = 195$, $N_2 = 442$, $N_3 = 180$), involving various external measures of empathy, general personality domains, emotion recognition, and other constructs, the total German Emotional Contagion Scale demonstrated sound convergent and discriminant validity. A bi-factor model provided acceptable fit, suggesting the factorial validity of the total scale, which is aimed to measure a general factor, representing the susceptibility to emotional contagion. Longitudinal analyses across four measurement occasions revealed high temporal stabilities for the total scale across periods of up to 1 year as well as longitudinal measurement invariance of the factor loadings and partial invariance of the intercepts and residuals. The correlation pattern of the mimicry short version was comparable to the total Emotional Contagion Scale's correlation pattern, the unidimensional factor structure was confirmed, and it also demonstrated high temporal stabilities and longitudinal invariance. The present research underscores the relevance of susceptibility to emotional contagion and mimicry as personality constructs and provides valid measurement tools for assessing them in future research and practical contexts (e.g., assessment in the clinical or work context).

## Introduction

Social interactions are fundamentally dependent on empathy, especially the ability to perceive and resonate with the emotions of others. One of the core components of empathy is emotional contagion – the nuanced process through which emotions are transmitted from one individual to another, thereby influencing the emotional dynamics of relationships and effective interpersonal functioning in general. Particularly, the interpersonal process of emotions spreading across humans (emotional contagion)

**Data availability statement:** All relevant data are within the manuscript and its Supporting Information files.

**Funding:** This work was supported by Grant 02L14A150 from the Federal Ministry of Education and Research (BMBF). The BMBF was not involved in study design, data collection, data analysis, data interpretation and in the writing of this report.

**Competing interests:** The authors have declared that no competing interests exist.

can be distinguished from an individual's tendency to emotionally resonate with other people (susceptibility to emotional contagion). To understand the latter construct, on whose measurement the present research is focusing, a definition of the former is at first crucial.

Emotional contagion can be defined as an interpersonal process, in which emotions felt and expressed by a person are transmitted to another person [1–3]. With respect to the mechanisms of emotional contagion, the, to our knowledge, most elaborate theory [2] describes three stages: *Mimicry*, *feedback*, and *contagion*. According to Hatfield and colleagues [2], mimicry describes the automatic synchronization of, for example, facial expressions, vocalizations, or postures between two interacting people. Second, the mimicry of expressions and behaviors induces a synchronized emotional state *via* (e.g., facial) *feedback* mechanisms [2]. Third, the observer's subjective emotional experience synchronizes with the other person's experience in response to these feedback mechanisms, labeled *contagion* [2,4]. However, in contrast to this theory – that views mimicry as a basic and general automatic tendency – other perspectives specifically emphasize the moderating contextual and social influences on mimicry [5]. According to this view, whether mimicry occurs between two persons substantially depends on interpersonal variables such as liking or familiarity [6]. Moreover, the latter perspective stresses the idea that mimicry does not refer to merely *copying* another's facial muscular movement, but involves the *interpretation* of facially displayed signals (e.g., instead of just recognizing a movement of the lips, people see a smile and interpret it as an expression of happiness) [5].

The frequent occurrence of emotional contagion in social interactions has been well documented (for an overview, see [4,7]). The phenomenon of emotional contagion receives attention from various disciplines. For instance, recent neuroscientific research demonstrated that emotional contagion for different emotions (e.g., happy and sad) is systematically related to different brain activation patterns [8]. Apart from that, several studies investigated the relations between emotional contagion and the synchrony of physiological parameters, such as heart rate or heart rate variability [9,10]. Apart from that, a recent study investigated emotional contagion in the context of political psychology, pointing out the significant transmission of politicians' emotions to observing participants [11]. Specifically, the latter study demonstrated that this effect was moderated by whether the politician belonged to the party the participant would vote for compared to another party.

### Emotional contagion and empathy

Emotional contagion can be considered a component of *empathy* [4], a term that has been both intensively and inconsistently addressed by psychological research (e.g., [12]). Even though the concept has been in the focus of scientific research since more than a century, a consent regarding its conceptualization or measurement remains absent, resulting in a glaring variety of definitions and measurements tools, complicating comparisons between studies [13–17]. Apart from all conceptual ambiguities, empathy can basically be divided into an *affective* and a *cognitive* component, which both include rather fundamental phenomena and higher-order processes

[12,18]: The former include *emotion sharing* (affective)*,* i.e., feeling what others are feeling, and *emotion recognition* (cognitive), i.e., detecting what others are feeling [12]. Higher order empathy processes include the affective process *empathic concern*, i.e., "feelings of warmth, compassion and concern for others undergoing negative experiences" ([19], p. 90) or the cognitive process *perspective taking*, i.e., adopting another's mental perspective. In order to integrate SEC into the complex empathy framework, it can be regarded as a basic affective empathic process [18], preceding higher order affective phenomena, such as empathic concern. The difference between the empathic subprocess of emotion sharing and emotional contagion can be found in the mechanism of both processes: As explained, emotional contagion is commonly conceptualized as an automatic process of getting "infected" with another's emotions, potentially caused by afferent feedback generated by mimicry [2]. In contrast, the affective-empathic process of sharing others' emotions can be deemed to presuppose several (e.g., cognitive) operations, such as perceiving another's state and mentally modelling his/ her situation – which in turn leads to a congruent emotional state as a result of understanding, e.g., the painful situation of the other person [17]. Another defining aspect of (affective) empathy is the distinction between one's own situation and feelings and the other person's, which is called self-other differentiation [12], delineating affective-empathic emotion shar- ing from emotional contagion as well.

Taken together, emotional contagion can be delineated from other components of empathy by its basal and automatic conceptualization that does not presuppose cognitive operations, such as recognizing, understanding, or modeling the other person's situation, feelings or thoughts. Therefore, it seems advisable for psychological assessment to focus on emotional contagion as an own basically relevant phenomenon instead of merely additionally addressing it within the scope of a broader measurement of empathy.

## Susceptibility to emotional contagion

Beyond the above discussed studies that focus on emotional contagion as an interpersonal process, another line of research investigates interindividual differences in the tendency to emotionally resonate with others – a personality trait that can be labeled *susceptibility to emotional contagion* (SEC) [2,20]. Specifically, SEC can be defined as "the tendency to automatically mimic and synchronize expressions, vocalizations, postures, and movements with those of another per- son's and, consequently, to converge emotionally" [21]. The tendency to express mimicry can therefore be considered a subconstruct of SEC, just as mimicry can be deemed a subprocess of emotional contagion.

SEC has been measured via different self-report instruments (for an overview of measures, see Marx) [22]. Most of these established questionnaires focus on other constructs, mostly the broader construct of empathy (for a comparison: see below) and only include a subscale measuring SEC [22], for example see the Emotional Empathic Tendency Scale [23]. In contrast, the Emotional Contagion Scale [20] specifically and explicitly focuses on SEC.

## The Emotional Contagion Scale

The Emotional Contagion Scale (ECS) is a 15-item questionnaire introduced by Doherty [20] and the – to our knowledge – most frequently administered instrument for measuring SEC. The ECS addresses the contagion of fear, love, anger, sad- ness, and happiness with three items each. An example item is "Being with a happy person picks me up when I'm feeling down." (item 02, happiness). The full scale can be found in the supporting information (S6 in S1 File).

**Factor structure.** The initial validation study for the ECS [20] reported a unidimensional structure. A German version of the ECS was published by Falkenberg [24], but, to our knowledge, this version has not yet been analyzed with respect to its psychometric properties, reliability, or validity. By contrast, Lundqvist [25] reported the extraction of three or five components as suggested by the scree plot within principal component analysis (PCA) for the Swedish version of the ECS. In Confirmatory Factor Analyses (CFA), he demonstrated that a unidimensional solution was not tenable, whereas models entailing five factors achieved the best fit. The five factors clustered the items on the basis of the specific emotions addressed by the content of each item, namely, fear, love, anger, sadness, and happiness (henceforth referred to as

*FLASH*). Apart from that, the PCA conducted by Kevrekidis et al. (Greek ECS) [26] suggested the extraction of either two or four components. After removing the items addressing fear (due to cross-loadings), forcing the items to load onto four components resulted in a factor loading pattern similar to the FLASH model (without the fear factor). Moreover, Lundqvist and Kevrekidis (Greek version) [27] confirmed the superiority of the FLASH model over a two-factor structure as well as a unidimensional solution, which both fitted poorly to the data. Besides that, a model comprising one (or two) second-order factor(s) instead of the correlations between the five factors reached a good (though significantly worse, according to $\chi^2$) fit as well, replicating Lundqvist [25].

In a series of CFA, Lo Coco et al. [28] demonstrated that the one-factor model was substantially outperformed by the FLASH model for the Italian version of the ECS. A (more parsimonious) four-factor model (fear and anger items loading onto a single factor) explained the data equally well. Second-order models also fitted the data well, while the best fit was achieved by models combining the four or five factors with a bi-factor. This pattern of results was replicated in a second study [28]. Eventually, Wrobel and Lundqvist [29] replicated the superiority of the FLASH model over the one- and two-factor models, which fitted the (Polish) ECS data poorly. Moreover, second-order models again yielded a good fit. These results call into question the factorial validity of the ECS, which is aimed at measuring a unidimensional construct. Notwithstanding, all of these validation studies also demonstrated the reasonable fit of a model that included a general second-order factor or a bi-factor model, suggesting that the ECS measures a unidimensional construct, while additionally indicating an influence of errors on the item responses. As Lo Coco et al. [28] pointed out, their results on a bi-factor model suggested that the response to each item is influenced by a general trait factor (SEC) and an emotion-specific factor. For instance, an item addressing the experience of sadness (e.g., Item 01) [20] is likely additionally affected by factors other than SEC (e.g., neuroticism). According to test theory, of course every item response is composed of a true score as well as an error term. However, the ECS in particular contains groups of items that share the same error influence because all the items in a group are related to the same emotion (e.g., Items 01, 04, and 14 are all related to the emotion of sadness), attenuating the fit of the one-factor model. To sum up, a consistent finding across previous validation studies is that the simple one-factor model is not sufficient for describing the dimensionality of the ECS. Nevertheless, in addition to theoretically weakly justified multidimensional models (e.g., FLASH), another way to deal with this issue involves models that entail a general factor but also account for emotion-specific influences (e.g., *via* first-order factors).

**Convergent and discriminant validity.** The initial validation study of the ECS [20] demonstrated a large positive association between the ECS and the Emotional Empathic Tendency Scale [23] as well as moderate associations with the (rather affective) subscales *empathic concern* and *personal distress* and small relationships with the (rather cognitive) subscales *perspective taking* and *fantasy* from the Interpersonal Reactivity Index [19]. Subsequent validation studies reported large and moderate associations between the ECS and the Interpersonal Reactivity Index subscales *empathic concern* and *perspective taking*, respectively [28,29]. This pattern of results is consistent with the (above explained) conceptualization of SEC as a fundamental affective empathic process, thereby arguing for the construct validity of the ECS. This interpretation was supported by a large correlation [30] between the ECS and the Toronto Empathy Questionnaire, which also captures empathy primarily as an affective phenomenon [16].

In addition, Doherty [20] demonstrated that the original English ECS was moderately positively associated with an index quantifying the self-reported congruent emotional response after being confronted with video stimuli depicting a person expressing happiness or sadness. Furthermore, its initial validation study as well showed that the ECS moderately positively correlated with *cue-responsiveness*, which indicates the extent to which a person reports a congruent subjective emotional experience after adopting a natural happy or sad facial expression. These findings highlight the convergent validity of the ECS.

Wrobel and Lundqvist [29] showed that the ECS was moderately positively associated with neuroticism and agreeableness, whereas it was uncorrelated (openness and conscientiousness) or only weakly correlated (extraversion) with the other Big Five personality traits, arguing for the convergent and discriminant validity of the scale, respectively. To our

knowledge, no information on the convergent or discriminant validity of the German version of the ECS is available to date.

**Content examination of the ECS items.** The ECS contains items tapping the basic process of mimicry (e.g., "If someone I'm talking with begins to cry, I get teary-eyed"; Item 01) as well as items addressing a congruent emotional experience (i.e., contagion, e.g., "Being with a happy person picks me up when I'm feeling down"; Item 02). With respect to the type of emotion, the ECS addresses the mimicry and contagion of fear, love, anger, sadness, and happiness with three items each. The selection of these concepts indeed does not report a commonly reported model (like Ekman's) [31]. Furthermore, discrete emotion models in general have been criticized, e.g., in light of lacking evidence for a specific correspondence between individual emotions and the activation of certain brain areas or dissociable neural interaction patterns (for an overview, see Barrett [32], for a reply see Colombetti) [33]. Moreover, the ECS items measuring fear overlap with the concept of stress, and some items addressing anger could be considered indicators of tension alike. Most importantly, a third limitation of the scale pertains to the items addressing love: It is noteworthy that Doherty [20] referred to Fisher [34] to provide a reason for why *love* should be considered a basic emotion. However, labeling love as an emotion itself instead of a complex interpersonal phenomenon involving several emotions is a controversial practice [35]. Moreover, the items addressing love [6,9,12] obviously do not capture a *contagious* phenomenon at all: For instance, a person could (honestly) agree with the statement "When I look into the eyes of the one I love, my mind is filled with thoughts of romance" (Item 06), even if the other person does not return these romantic thoughts.

## The present studies

First, the present studies were aimed at validating the German version of the ECS, including its factor structure, convergent and discriminant validity, longitudinal measurement invariance, and temporal stability. In addition, given that, to the best of our knowledge, no brief version of the ECS is yet available, the second aim of the present research was to develop and validate a shorter version of the ECS. The most established conceptualization of SEC particularly emphasizes its characteristic of being a basal and automatic phenomenon [2]. Therefore, we focused, in the short scale, on the items explicitly tapping mimicry – the most basic subprocess of SEC, which is confounded with the phenomenon of a congruent subjective emotional experience in the total scale. The newly developed shorter scale is thereby aimed at consequently implementing the conceptualization of SEC as a *primitive* phenomenon {Hatfield, 1992 #3425} by capturing the most basic (and overt) subprocess of SEC (mimicry) specifically, which could represent a promising access to the broader construct of SEC. An analogy may be drawn to Raven's matrices, which are – initially created as a measure of the eduction of relations and correlates – often used as a measurement approach to the *g* factor of intelligence (for an overview, see, e.g., Mackintosh & Bennett) {Mackintosh, 2005 #3989}. In Study 1, we focused on the dimensionality and the convergent and discriminant validity of the total German ECS and the development and validation of the mimicry brief version.

Study 2 expanded the dimensionality and convergent and discriminant validity analyses of both ECS versions. Particularly, Study 2 also investigated the longitudinal measurement invariance and temporal stability of both scales across periods of up to 1 year.

Finally, in Study 3 we aimed to replicate and extend the results of Studies 1 and 2 on the factor structure and the convergent and discriminant validity of both ECS versions. Apart from that, Study 3 also addressed an issue associated with the wording of a specific item [03] by testing the psychometric effects of a reformulation.

## Study 1

In Study 1, we analyzed the dimensionality, internal consistency, and convergent and discriminant validity of the German adaptation of the ECS. Moreover, we developed and validated a brief mimicry version of the ECS. To address the issues discussed above regarding the items addressing the contagion of love (Items 06, 09, and 12), we decided to exclude these items from the German version of the ECS. However, results for a version of the scale including these three items

can be found in the Supporting information (S2 in S1 File). To avoid confusion, we will, in the following, refer to the 12-item ECS version (without the love items), which is analyzed in the present research, by the term *susceptibility to Emotional Contagion Scale (sECS)*. As explained in the Introduction, previous results on the factor structure of the ECS are compatible with the conceptualization of SEC as a unidimensional trait, while pointing out an additional emotion-specific influence on item responses. To address this issue, we tested a model comprising one factor and correlated error terms for the items tapping the same emotion, namely, sadness (Items 01, 04, 14), happiness (Items 02, 03, 11), anger (Items 05, 07, 10), and fear (Items 08, 13, 15) as well as a (more complex) bi-factor model (as suggested by Lo Coco et al.) [28] in addition to the one-factor model (as originally proposed by Doherty) [20]. The three models are depicted in Fig 1. We expected the one-factor model to show weak fit to the data from the total sECS scale and to be outperformed by the model with correlated error terms. Likewise, we expected the more complex bi-factor model, which accounted for relationships between the emotion-specific factors, to show the best fit (as also reported by Lo Coco et al.) [28].

### Development of a mimicry brief version of the sECS

In addition, we developed a mimicry brief version of the sECS consisting of Items 01, 03, 05, and 13 (labeled *sECS-mimicry*). These items were theoretically selected for two reasons: On the one hand, they capture *mimicry*, the first stage of SEC (as described above), which can be considered to reflect SEC in the most basic way [2]. Moreover, this first stage of Hatfield's theoretical model refers to manifest, overt behavior, whereas the third stage implies latent inner states (*feelings*) that cannot be observed directly and are presumably difficult to evaluate – even for the person who is experiencing them. By contrast, in the total sECS, the four items tapping mimicry are blended with items tapping the latent contagion stage.

On the other hand, and to ensure that the brief version broadly captures the construct, we carefully observed that the emotions sadness, happiness, anger, and "fear" were addressed with one item each. Although labeled as addressing the contagion of fear [20], Item 13 on the total sECS (i.e., Item 04 on the sECS-mimicry) might be more appropriately considered an indicator of the contagion of stress. For the sECS-mimicry, a one-factor model was tested.

The language adaptation process was conducted in accordance with the *ITC Guidelines for Translating and Adapting Tests* as published by the International Test Commission [36]. Specifically, to test the quality of the German translation, we used a professional translator (native English speaker but fluent in the German language) who translated all the German sECS items back into the English language. Then another professional translator (native English speaker) compared the original and the back-translated English versions with each other. No relevant differences were found, and thus, the initial German translation [24] was approved.

### Convergent and discriminant validity

In terms of convergent validity and in line with previous studies (see the Introduction), we expected positive associations between the total sECS and the sECS-mimicry and the Interpersonal Reactivity Index [19], the Basic Empathy Scale (BES) [37], as well as neuroticism and agreeableness. Furthermore and on the basis of the theory that there is a link between the construct *empathy* – which we conceptualize as including SEC (see above) – and *altruism* (e.g., [38]), we expected positive correlations between both versions of the sECS and self-reported altruism. In terms of discriminant validity, we hypothesized no or small (sensu Cohen) [39] associations between both sECS versions and conscientiousness, extraversion, and openness as well as with a measure of socially desirable responding [40].

### Method

**Participants and data collection.** The data were collected *via* an online survey as part of a project focusing on a validation of the German version of another questionnaire (Toronto Empathy Questionnaire) [41]. Participants were primarily students at the university of Duisburg-Essen and were compensated for their participation with partial course credit or the opportunity to participate in a raffle. Data were collected between April, 13, 2022, and June 07,

## One Factor

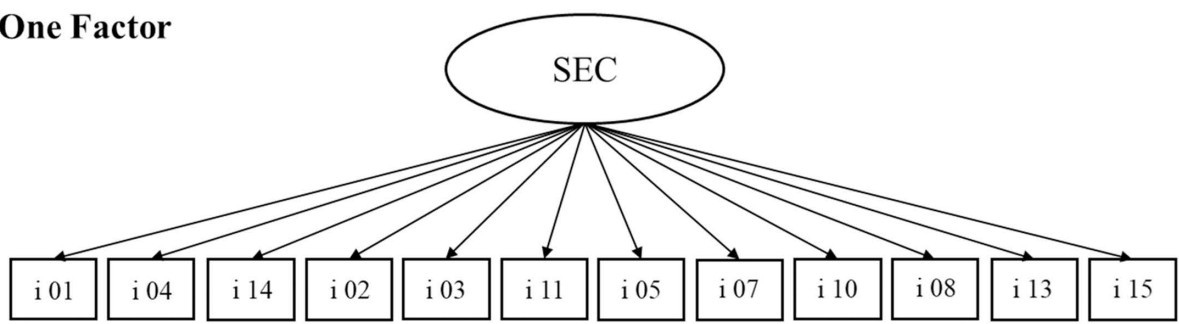

## One Factor with Correlated Errors

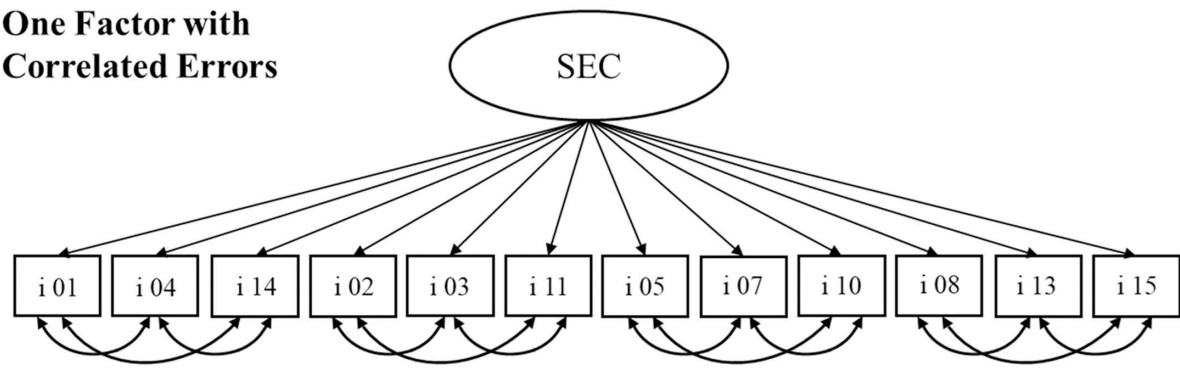

## Bi-Factor

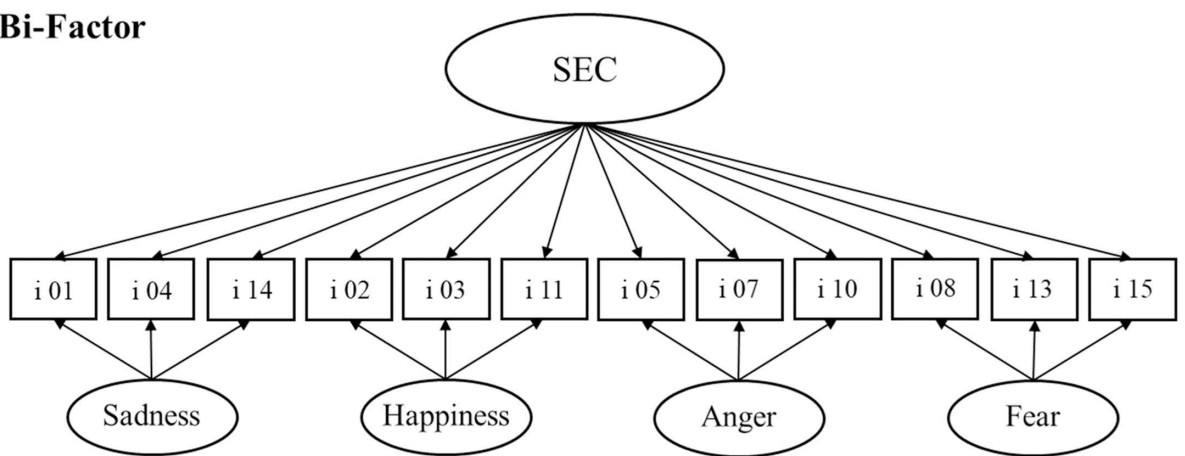

**Fig 1. Measurement models for the sECS.** SEC = susceptibility to emotional contagion; i = item.

2022. Participants provided written informed consent. For the correlational analyses, we aimed to recruit at least 153 participants, which represented the number of participants needed to find a correlation of about *r* = .20 with an alpha level (one-tailed) of .05 and a power of .80 (computed by G*Power) [42]. For the factor analyses, we aimed to recruit at least 187 participants. This was the number of participants required to test a model with 42 degrees of freedom (one-factor model

with correlated error terms) with an RMSEA of about .06, an alpha level of .05, and a power of .80 (computed by semPower) [43]. A total of 195 participants (80.5% women) aged between 18 and 62 years ($M$ = 23.0, $SD$ = 6.6; one missing value) completed all measures (incomplete responses were deleted). Of the participants, 93.3% were students, 4.6% were employed, 1.0% were unemployed and 1.0% were retired.

**Measures.** The internal consistencies and descriptive statistics of external measures can be found in Tables 2 and 3, respectively. The consistency and descriptive statistics of the sECS and sECS-mimicry can be found in the results section.

*sECS.* The Emotional Contagion scale was introduced by Doherty [20] to assess SEC, originally comprising 15 items. In the present research, the *love* items (Items 06, 09, and 12) were excluded from the computation of the total score (*sECS*). The original English items were translated into German by Falkenberg [24]. A four-point Likert scale (1 = *never*, 2 = *rarely*, 3 = *often*, and 4 = *always*) as initially chosen by Doherty [20] was used. In addition, we examined a mimicry brief version of the sECS: The *sECS-mimicry* contains the original [20] Items 01, 03, 05, and 13. For both the total sECS and the sECS-mimicry, the sum of the item scores was used in the analyses. An example item is "If someone I'm talking with begins to cry, I get teary-eyed." (item 01; for all items see S6 in S1 File).

*Interpersonal Reactivity Index*. The Interpersonal Reactivity Index [19] is a frequently used measure of empathy [12]. It comprises two rather affective subscales, namely, *empathic concern* (example item: "I am often quite touched by things that I see happen.") and *personal* distress (e.g., "In emergency situations, I feel apprehensive and ill-at-ease."), as well as the two rather cognitive subscales *perspective taking* (e.g., "I believe that there are two sides to every question and try to look at them both.") and *fantasy* (e.g., "I really get involved with the feelings of the characters in a novel."). The present research administered the German version as introduced by Paulus [44], consisting of 16 (positively scored) items. Responses were given on a five-point Likert scale (1 = *never*, 2 = *rarely*, 3 = *sometimes*, 4 = *often*, and 5 = *always*).

*Basic Empathy Scale*. The Basic Empathy Scale, introduced by Jolliffe and Farrington [37], German version by Heynen et al. [45] is a self-report instrument consisting of an affective (e.g., "After being with a friend who is sad about something, I usually feel sad.") and a cognitive (e.g., "I can understand my friend ' s happiness when she/he does well at something.") empathy subscale. The items address different emotions, namely, *anger*, *fear*, *sadness*, and *happiness*. Responses were given on a five-point Likert scale (1 = *I don't agree*, 2, 3, 4, 5 = *I fully agree*).

*Facets of Altruistic Behaviors Scale.* The Facets of Altruistic Behaviors Scale [46] measures different facets of altruistic behavioral traits. We administered only the five-item subscale *help giving* (HG), which captures the propensity to share one's resources with needy or deserving others (e.g., "I would certainly jeopardize my own well-being in order to help hungry and sick people."). Responses were given on a six-point Likert scale (1 = *strongly disagree*, 2, 3, 4, 5, 6 = *strongly agree*).

*NEO-Five-Factor Inventory-30.* The NEO-Five-Factor Inventory-30 [47] is a 30-item version of the NEO-five-factor inventory by Costa & McCrae [48], German version by Borkenau and Ostendorf [49] for assessing the Big Five personality traits with six items per dimension (e.g., "I like having lots of people around me", extraversion). The response format was a five-point Likert scale (1 = *doesn't apply*, 2, 3, 4, 5 = *applies*).

*Short Scale Social Desirability-Gamma.* The Short Scale Social Desirability-Gamma [40] is a brief measure of socially desirable responding. It comprises the two subscales *exaggeration of positive qualities* (e.g., "When I talk to someone, I always listen to them carefully.") and *understatement of negative qualities* (e.g., "It has happened before that I have taken advantage of someone.") with three items each. The items were presented interspersed within the NEO-five-factor inventory-30 and thus used the same response scale.

### Results and discussion

**Confirmatory factor analyses.** Confirmatory factor analyses (CFA) were performed with the R package *lavaan* [50] using the Maximum Likelihood estimator. Due to a violation of the multivariate normality assumption of the maximum likelihood estimator for all items in all studies, all analyses in all studies were repeated using a Satorra-Bentler (SB)

$\chi^2$ correction [51]. Due to the limited number of response categories (four) in Study 1, the CFA was repeated using the Diagonally Weighted Least Squares (DWLS) estimation method (as suggested by Mindrila [52]; see Supporting information, S1 in S1 File).

For determining the absolute fit of each model, the Root Mean Square Error of Approximation (RMSEA) and the Standardized Root Mean Square Residual (SRMR) were considered, while the Comparative Fit Index (CFI) and the Tucker-Lewis Index (TLI) were computed to examine the incremental fit of each model. According to established recommendations [53,54], RMSEA (SRMR) values less than.06 (.08) were taken to indicate sufficient model fit, while CFI and TLI values higher than.90 and.95 were interpreted as reflecting acceptable and excellent model fit, respectively. For model comparisons, we employed as well the $\chi^2$ difference test as the Akaike Information Criterion (AIC). Differences in AIC > 10 were deemed to reflect an essentially large effect [55]. In case of discrepancies between $\chi^2$ difference and AIC, we deem AIC the more relevant criterion in light of the sample size sensitivity of $\chi^2$ [56]. Table 1 presents the results. For the sECS, the one-factor model did not fit the data well. According to the (SB-corrected) $\chi^2$ differences, the model with one factor and correlated errors fitted the data significantly better, $\Delta\chi^2(\Delta df = 12) = 130.711$, $p < .001$, but was likewise outperformed by the bi-factor model, $\Delta\chi^2(\Delta df = 6) = 38.087$, $p = .009$. Both comparisons were clearly corroborated by the AIC values. While the model with one factor and correlated errors did not fit the data well with respect to both absolute and incremental indices, the bi-factor model showed excellent fit according to CFI, RMSEA, and SRMR and acceptable fit according to TLI. However, the TLI of the bi-factor model was below Hu and Bentler's [53] cutoff for excellent model fit (.95, see above). For the sECS-mimicry, the fit of the one-factor model was acceptable only according to the SRMR, whereas the other indices indicated poor fit.

Taken together, the one-factor model and the model with one factor and correlated errors clearly showed misfit to the data. The bi-factor model showed excellent fit only according to the indices except for the TLI. The one-factor model for the sECS-mimicry could not be confirmed in Study 1. Nevertheless, in light of the sample size, which was below 250, especially the TLI and RMSEA may have been too restrictive [53]. We addressed this limitation in terms of limited sample size for CFA in Study 2.

**Reliability.** The sECS reached values of $\alpha = .69$ (SE = .03) and $\omega = .77$ (SE = .03), whereas the internal consistency of the sECS-mimicry was $\alpha = .44$ (SE = .06) and $\omega = .54$ (SE = .10). Ordinal $\alpha$ and $\omega$ were also computed: The sECS achieved an ordinal $\alpha$ of.74 (SE = .03) and an ordinal $\omega$ of.75 (SE = .03). The sECS-mimicry reached an ordinal $\alpha$ of.48 (SE = .07) and an ordinal $\omega$ of.51 (SE = .06). Taken together, the internal consistency was close to the threshold for evaluating individual differences (cutoff: $r_{tt} \geq .70$) for the sECS and close to the threshold for evaluating group differences (cutoff: $r_{tt} \geq .50$) for the sECS-mimicry [57]. The consistency of both scales is thus lower than expected, which could have been caused by several factors: Besides the limited sample size (as already mentioned), the response scale may have been problematic: A four-point Likert does not only contain a limited number of options (only two besides the extreme categories), potentially restricting the (substantial) variance in the data. It also prohibits the participants from responding neutrally to any question, thus forcing a truly neutral response being converted into a high or low response, potentially increasing the error variance

**Table 1. CFA results for Study 1 (Satorra-Bentler corrected test statistic in parentheses).**

| Scale | Model | $\chi^2$ | df | CFI | TLI | RMSEA | SRMR | AIC |
|---|---|---|---|---|---|---|---|---|
| sECS | 1F | 222.666 (204.893) | 54 | .524 (.537) | .419 (.434) | .127 (.120) | .110 (.110) | 5269.249 |
| | 1F + CE | 91.955 (84.412) | 42 | .859 (.870) | .779 (.795) | .078 (.072) | .074 (.074) | 5162.538 |
| | BF | 53.868 (50.543) | 36 | .950 (.955) | .908 (.918) | .050 (.046) | .051 (.051) | 5136.451 |
| sECS-mimicry | 1F | 7.194 (6.053) | 2 | .845 (.860) | .536 (.581) | .115 (.102) | .050 (.050) | 1747.204 |

*Note.* 1F = One-factor model; 1F + CE = Model with one factor and correlated errors; BF = Bi-factor model. Shapiro-Wilk Tests indicated deviations from normality for all items.

in the data, which could decrease consistency [58]. Third, the content heterogeneity of both scales (especially the sECS-mimicry), which tap different emotions, could decrease consistency [59,60]. In the face of content heterogeneity, a low internal consistency should not necessarily be deemed an indicator of a lack of a scale's psychometric quality [59,60], especially in light of high test-retest reliability [61]. In fact, previous studies demonstrated test-retest reliabilities for the ECS almost as high as its internal consistency [20,29] or even higher [25]. Apart from that, Participants' scores on the sECS ranged between 21 and 44, $M = 34.6$, $SD = 4.5$, while the sECS-mimicry scores ranged between 6 and 15, $M = 10.9$, $SD = 1.9$.

**Convergent and discriminant validity.** The correlation between the sECS and the sECS-mimicry was $r = .83$. The correlations of the sECS (sECS-mimicry) and the external measures can be found in Table 2 (Table 3). The significance level was adjusted according to the false discovery rate (FDR) [62]. The correlations were largely in line with our hypotheses: Both sECS scales significantly positively correlated with the empathy measures (exception: Perspective Taking subscale of the Interpersonal Reactivity Index for the sECS-mimicry). The positive correlation to altruism was only found for the sECS, but not for the sECS-mimicry and even for the sECS, it was only marginal – the lower border of the 95% confidence interval was .02, which is only slightly deviant from zero, limiting the practical relevance of this finding. Third, we found significant (though weak) positive associations with conscientiousness (both scales) and openness to experience (only sECS), which were against expectations.

Taken together, we generally confirmed the predicted pattern of correlations for the sECS, arguing for the convergent and discriminant validity of the scale. Moreover, the correlation pattern of the sECS-mimicry was very similar to the associations found for the total scale. This finding is remarkable given that the scale has only four items, thus attesting to the validity of the brief mimicry version as well.

**Table 2. Internal consistencies for external measures, correlations between the sECS and external measures of Study 1.**

| Measure – Scale | α (SE) | ω (SE) | Correlation to sECS | | | | |
|---|---|---|---|---|---|---|---|
| | | | r | lb | ub | p | α (fdr) |
| IRI – EC | .58 (.05) | .63 (.08) | .46 | .34 | .56 | **<.001** | .007 |
| IRI – PT | .79 (.03) | .84 (.03) | .22 | .08 | .35 | **.001** | .021 |
| IRI – FT | .68 (.04) | .79 (.03) | .37 | .24 | .48 | **<.001** | .011 |
| IRI – PD | .80 (.03) | .82 (.16) | .36 | .24 | .48 | **<.001** | .014 |
| BES – AE | .64 (.05) | .79 (.04) | .61 | .51 | .69 | **<.001** | .004 |
| BES – CE | .78 (.03) | .85 (.03) | .32 | .19 | .44 | **<.001** | .018 |
| FAB – HG | .71 (.03) | .75 (.06) | .16 | .02 | .29 | **.015** | .036 |
| NEO – A | .69 (.03) | .80 (.03) | .20 | .06 | .33 | **.003** | .025 |
| NEO – C | .79 (.02) | .85 (.02) | .16 | .02 | .3 | **.011** | .032 |
| NEO – E | .75 (.03) | .86 (.02) | .12 | −.03 | .25 | .054 | .043 |
| NEO – N | .86 (.02) | .91 (.02) | .19 | .06 | .33 | **.003** | .029 |
| NEO – O | .82 (.02) | .86 (.02) | .13 | −.01 | .27 | **.031** | .039 |
| KSE-G – PQ+ | .47 (.07) | .48 (.07) | −.01 | −.15 | .13 | .575 | .05 |
| KSE-G – NQ- | .55 (.06) | .60 (.07) | .09 | −.05 | .23 | .111 | .046 |

*Note.* SE = Standard error. Lb (ub) = lower (upper) border of the 95% confidence interval of the correlation coefficient. α (fdr) = Adjusted significance level according to the false discovery rate. Interpersonal Reactivity Index = Interpersonal Reactivity Index; EC = empathic concern; PT = perspective taking; FT = fantasy; PD = personal distress; BES = Basic Empathy Scale; AE = affective empathy; CE = cognitive empathy; FAB = Facets of Altruistic Behaviors Scale; HG = help giving; NEO = NEO-Five-Factor Inventory-30; A = agreeableness; C = conscientiousness; E = extraversion; N = neuroticism; O = openness; KSE-G = Short Scale Social Desirability-Gamma; PQ+ = exaggeration of positive qualities; NQ- = understatement of negative qualities. All p values are one-tailed. Significant p values in bold. Shapiro-Wilk Tests indicated deviations from normality for all items.

**Table 3. Descriptives of external measures and correlations between the sECS-mimicry and external measures of Study 1.**

| Measure – Scale | Descriptives | | | | Correlation to sECS-mimicry | | | | |
|---|---|---|---|---|---|---|---|---|---|
| | M | SD | Min | Max | r | lb | ub | p | α (fdr) |
| IRI – EC | 34.6 | 4.5 | 21 | 44 | .36 | .23 | .48 | **<.001** | .007 |
| IRI – PT | 10.9 | 1.9 | 6 | 15 | .12 | −.02 | .26 | .041 | .032 |
| IRI – FT | 14.9 | 2.1 | 9 | 20 | .30 | .16 | .42 | **<.001** | .014 |
| IRI – PD | 14.3 | 2.7 | 7 | 20 | .33 | .2 | .45 | **<.001** | .011 |
| BES – AE | 14.6 | 2.7 | 7 | 20 | .55 | .45 | .64 | **<.001** | .004 |
| BES – CE | 12 | 3.3 | 5 | 20 | .28 | .14 | .4 | **<.001** | .018 |
| FAB – HG | 23.9 | 3 | 13 | 30 | .06 | −.08 | .2 | .205 | .043 |
| NEO – A | 24.3 | 3.1 | 14 | 30 | .14 | 0 | .27 | **.026** | .029 |
| NEO – C | 18.8 | 4.1 | 10 | 30 | .18 | .04 | .31 | **.005** | .021 |
| NEO – E | 23.8 | 3.8 | 14 | 30 | .06 | −.08 | .2 | .208 | .046 |
| NEO – N | 23.9 | 4.1 | 14 | 30 | .17 | .03 | .3 | **.01** | .025 |
| NEO – O | 19.4 | 4.1 | 9 | 29 | .12 | −.02 | .26 | .042 | .036 |
| KSE-G – PQ+ | 17.3 | 5.7 | 6 | 30 | −.02 | −.16 | .12 | .603 | .05 |
| KSE-G – NQ- | 21.1 | 5.4 | 6 | 30 | .08 | −.06 | .22 | .119 | .039 |

*Note.* SE = Standard error. Lb (ub) = lower (upper) border of the 95% confidence interval of the correlation coefficient. α (fdr) = Adjusted significance level according to the false discovery rate. Interpersonal Reactivity Index = Interpersonal Reactivity Index; EC = empathic concern; PT = perspective taking; FT = fantasy; PD = personal distress; BES = Basic Empathy Scale; AE = affective empathy; CE = cognitive empathy; FAB = Facets of Altruistic Behaviors Scale; HG = help giving; NEO = NEO-Five-Factor Inventory-30; A = agreeableness; C = conscientiousness; E = extraversion; N = neuroticism; O = openness; KSE-G = Short Scale Social Desirability-Gamma; PQ+ = exaggeration of positive qualities; NQ- = understatement of negative qualities. All *p* values are one-tailed. Significant *p* values in bold. Shapiro-Wilk Tests indicated deviations from normality for all items.

## Study 2

Study 2 addressed the shortcomings of Study 1, namely, the moderate sample size and the limited response format. Moreover, Study 2 focused on another group of participants, namely, nursing staff (instead of students, as commonly used). This different sample was chosen to ensure that the results of the present research are not based solely on one kind of homogenous and highly educated student sample, which could have undermined the generalizability of the findings. Nursing is a profession that is commonly regarded as being especially dependent on empathic processes and traits (e.g., see Brunero et al.) [63]. Therefore, nursing staff can be deemed an especially interesting group for scientifically studying empathy and associated concepts such as emotional contagion or emotion recognition both on the interpersonal level as in terms of the individual susceptibility for these processes. We therefore chose a nursing sample for Study 2, even though the gender imbalance (see Study 1) can unfortunately not be simultaneously addressed, given that nursing remains a profession dominated by female employees. With a larger cross-sectional sample, Study 2 was designed to replicate the CFA models tested in Study 1 and reexamine the internal consistencies of the sECS and sECS-mimicry. According to Doherty's [20] observation of an increase in internal consistency after (among other changes) adding a response option, we expected a higher internal consistency in Study 2 (which used a five-point Likert scale) compared with Study 1 (which used a four-point scale). In addition, Study 2 examined the psychometric properties of the sECS and sECS-mimicry and extended the convergent and discriminant validity analyses.

On the basis of Hatfield et al. [21], we expected a positive correlation between the sECS (and sECS-mimicry) and a behavioral-based measure of emotion recognition [64]. In addition, we aimed to replicate the correlational pattern found in Study 1 between both sECS versions and general personality domains (Big Five) by using another model of personality structure (HEXACO) [65]. On the basis of the Study 1 results, we expected positive associations between both

sECS-versions and emotionality and agreeableness but no associations with the other HEXACO dimensions. In Study 2, using a longitudinal sample, we also analyzed the temporal stability and longitudinal measurement invariance of the sECS and the sECS-mimicry across intervals of 3–4, 6–8, and 9–12 months.

## Method

**Participants and data collection.** The data collection was conducted as part of a joint project focusing on an evaluation of an intervention for nursing staff [66]. The intervention aimed to develop and maintain a functional approach to the empathic experience of caregivers and to contribute to their emotional relief. The intervention included both short-term training, e.g., psychoeducation and case examples, and longer-term coaching methods, e.g., communicating one's needs [66]. Data were collected at four measurement occasions, each spaced 3–4 months apart. Participants with more than three missing values in one instrument were excluded from the analyses. Individual missing values were replaced by item means. Data were collected between June 16, 2016, and July 16, 2018. Participants provided written informed consent.

For the present study, three subsamples were derived from the total sample: Sample 1 included 442 participants (81.3% women; 3 missing values) aged between 19 and 72 years ($M = 37.57$, $SD = 11.12$; 5 missing value) who completed the sECS and the HEXACO-60 [65] at the first measurement occasion. All participants were employed and 64.3% indicated their highest level of education as A-level or higher. Sample 2 contained 231 participants (of the 442 from Sample 1) who additionally completed the behavioral-based assessment of emotion recognition [64] at the first measurement occasion. Samples 1 and 2 were cross-sectionally examined to determine the convergent and discriminant validity of the sECS. Moreover, Sample 1 was used to determine the dimensionality, internal consistency, and psychometric properties of the sECS. Data were collected from Samples 1 and 2 before any intervention began, wherefore the later treated intervention group was not excluded from these samples. By contrast, for Sample 3, only the untreated control group was considered ($n = 134$). Sample 3 was used to establish the longitudinal measurement invariance and the temporal construct stability of the sECS across the four abovementioned measurement occasions. The demographic information of the three samples is summarized in Table 4. The study was approved by the Ethics Committee of the Department of Computer Science and Applied Cognitive Science of the Faculty of Engineering of the University of Duisburg-Essen.

**Measures.** *sECS.* We used the German adaptation as described in Study 1 (Falkenberg's translation without Items 06, 09, and 12), except that the response format was a five-point Likert scale (0 = *never*, 1 = *rarely*, 2 = *sometimes*, 3 = *often*, and 4 = *always*).

*HEXACO-60.* The HEXACO-60 [65,67] assesses the six dimensions of the HEXACO personality model with 10 items each. The dimensions (Cronbach's α in the present study in parentheses) comprise honesty-humility (.675), with an

**Table 4. Demographics of Study 2 samples.**

|  |  | Sample 1 | Sample 2 | Sample 3 |
|---|---|---|---|---|
| n |  | 442 | 231 | 134 |
| Gender [%] | Female | 81.3 | 84.2 | 82.1 |
|  | Male | 18.2 | 15.8 | 17.9 |
| Age [Years] | *M* | 37.57 | 38.68 | 39.04 |
|  | *SD* | 11.12 | 11.27 | 11.30 |
|  | Range | 19–72 | 21–72 | 20–61 |
| Education (Highest Level) [%] | High school | 1.4 | 0.4 | 0 |
|  | O-levels | 34.3 | 35.1 | 29.1 |
|  | A-levels/higher | 64.3 | 64.5 | 70.9 |

example item "I wouldn't use flattery to get a raise or promotion at work, even if I thought it would succeed.", emotionality (.722), e.g., "I would feel afraid if I had to travel in bad weather conditions.", extraversion (.678), e.g., "I feel reasonably satisfied with myself overall.", agreeableness (.649), e.g., "I rarely hold a grudge, even against people who have badly wronged me.", conscientiousness (.701), e.g., "I plan ahead and organize things, to avoid scrambling at the last minute.", and openness to experience (.716), e.g., "I'm interested in learning about the history and politics of other countries.". Responses were given on a five-point Likert scale (1 = *totally disagree*, 2, 3, 4, 5 = *totally agree*) and we computed scale scores based on means (instead of sums). Participants' scores on Honesty-Humility ranged between 2 and 5, $M$ = 3.6, $SD$ = 0.6. Participants' scores on Emotionality ranged between 1.4 and 4.5, $M$ = 3.1, $SD$ = 0.5. Participants' scores on Extraversion ranged between 1.6 and 4.7, $M$ = 3.6, $SD$ = 0.5. Participants' scores on Agreeableness ranged between 1.5 and 4.5, $M$ = 3.2, $SD$ = 0.5. Participants' scores on Conscientiousness ranged between 1.8 and 4.8, $M$ = 3.6, $SD$ = 0.5. Participants' scores on Openness ranged between 1.8 and 4.9, $M$ = 3.4, $SD$ = 0.6.

*Geneva Emotion Recognition Test (Short).* The short version of the Geneva Emotion Recognition Test [64] is a behavioral-based measurement approach to emotion recognition. It comprises 42 brief video clips with actors/actresses expressing 14 different emotions. After each video clip, the participants were asked to choose which of the 14 emotions was expressed. Responses were scored 1 (*correct*) or 0 (*incorrect*). The internal consistency in the present study was α = .658. In addition, the split-half reliability was examined using the odd-even method (assigning every odd item to the first half of the test and every even item to the second). The correlation between the two halves was $r$ = .519, 95% CI [.418,.608], $p$ < .001. The Spearman-Brown-corrected split-half reliability was $r$ = .683. Participants' scores on the test ranged between 6 and 37, $M$ = 25.16, $SD$ = 5.02.

## Results and discussion

**CFA.** CFA were conducted in the same manner as in Study 1. As can be gathered from Table 5, the one-factor model again showed weak fit to the sECS data. By contrast, the model with one factor and correlated error terms fitted the data significantly better, $\Delta\chi^2(\Delta df = 12)$ = 337.805, $p$ < .001, but was likewise outperformed by the bi-factor model, $\Delta\chi^2(\Delta df = 6)$ = 39.366, $p$ = .015. These model comparisons were strongly corroborated by the AIC values. The model with one factor and correlated error terms achieved excellent fit with respect to CFI, RMSEA and SRMR, while the TLI was lower than Hu and Bentler's [53] cutoff for excellent model fit (.95). The bi-factor model demonstrated excellent fit to the data. The one-factor model tested for the sECS-mimicry showed excellent model fit according to CFI, RMSEA, and SRMR, while the TLI value was > 1. The findings point out a mathematical peculiarity of the TLI, which is not normed to necessarily lie between zero and one, wherefore it is also called *Non-Normed Fit Index* (NNFI) [68], but could also indicate overfit of the model, likely due to the small number of degrees of freedom.

**Reliability and psychometric properties.** The internal consistency of the sECS was α = .77 (SE = .02), ω = .82 (SE = .01). The internal consistency of the sECS-mimicry was α = .49 (SE = .04), ω = .53 (SE = .09). Ordinal α and ω were

**Table 5. CFA results for Study 2 (Satorra-Bentler corrected test statistic in parentheses).**

| Scale | Model | $\chi^2$ | $df$ | CFI | TLI | RMSEA | SRMR | AIC |
|---|---|---|---|---|---|---|---|---|
| sECS | 1F | 433.681 (391.566) | 54 | .645 (.627) | .566 (.544) | .126 (.119) | .098 (.098) | 13487.520 |
| | 1F+CE | 95.876 (88.737) | 42 | .950 (.948) | .921 (.919) | .054 (.050) | .047 (.047) | 13173.716 |
| | BF | 56.510 (52.336) | 36 | .981 (.982) | .965 (.967) | .036 (.032) | .032 (.032) | 13146.350 |
| sECS-mimicry | 1F | 1.275 (1.172) | 2 | >.999 (>.999) | 1.025 (1.033) | <.001 (<.001) | .014 (.014) | 4434.644 |

*Note.* 1F = One-factor model; 1F+CE = Model with one factor and correlated errors; BF = Bi-factor model. Shapiro-Wilk Tests indicated deviations from normality for all items.

also computed: The sECS achieved an ordinal α of .80 (SE = .02) and an ordinal ω of .80 (SE = .02). The sECS-mimicry reached an ordinal α of .53 (SE = .04) and an ordinal ω of .55 (SE = .04).

Taken together, we found that the internal consistency increased slightly in Study 2 (using a five-point Likert scale) compared to Study 1 (which used a four-point scale) for the sECS, but not (concerning ω) for the sECS-mimicry. The differences were small, wherefore the use of a four-point scale can – based on our present data – not be regarded a fundamentally bad decision. Nevertheless, the theoretical argument that an even number of response categories prohibits the participants from reporting truly neutrally, potentially producing error variance, since answers are artificially forced into high or low categories (see above) remains. Consistent to this idea, a recent meta-analysis reported higher reliability and validity of response scales with an odd number of categories compared to scales using an even number [58]. We conducted Item Response Theory (IRT) analyses to examine whether the neutral (i.e., middle) response category was used validly by the participants. Following Kankaraš {Kankaraš, 2025 #3987}, we implemented Partial Credit Models and then computed category characteristic curves (CCCs) for every response category and item, which show the relation between the probability of choosing one response category and the respondent's latent trait. As comparisons of the CCCs between the different response categories for every item (see S8 in S1 File) revealed, the CCC of the middle category was in each case distributed between those from the two adjoint response categories. These results suggest that the neutral category was used validly by the participants {Kankaraš, 2025 #3987}. We therefore decided to maintain the five-point Likert scale (which has recently been identified the most widely used in survey research) [58] as implemented in Study 2 for both versions of the sECS from now on.

Besides that, the sECS item means ranged between 0.87 and 3.31, while the SD ranged between 0.63 and 1.24. The detailed factor loadings, means, standard deviations, item-total correlations, and response probabilities of the items can be found in Table 6. Participants' scores on the sECS ranged between 7 and 43, $M = 27.6$, $SD = 6.1$, while the sECS-mimicry scores ranged between 2 and 14, $M = 8$, $SD = 2.2$. As can be gathered from Table 6, the items generally had high factor loadings ($λ > .30$) and item-total correlations ($rit > .40$), with the items 02, 03, and 11 being the only exceptions. These exceptions seem barely surprising in light of the positive valence of these items compared to the negative valence of all other items.

**Table 6. Psychometric properties of the sECS and sECS-mimicry in Study 2.**

| Item | $λ_{Total}$ | $λ_{Mimicry}$ | M | SD | $rit_{Total}$ | $rit_{Mimicry}$ | P |
|---|---|---|---|---|---|---|---|
| 01 | .323 | .419 | 1.55 | 0.98 | .48 | .41 | .39 |
| 02 | .145 | | 2.87 | 0.71 | .38 | | .72 |
| 03 | .155 | .162 | 3.25 | 0.63 | .41 | .27 | .81 |
| 04 | .490 | | 2.13 | 0.95 | .59 | | .53 |
| 05 | .526 | .594 | 0.87 | 0.94 | .51 | .52 | .22 |
| 07 | .554 | | 2.17 | 1.02 | .49 | | .54 |
| 08 | .500 | | 2.35 | 1.03 | .45 | | .59 |
| 10 | .599 | | 2.52 | 0.94 | .55 | | .63 |
| 11 | .143 | | 3.31 | 0.65 | .41 | | .83 |
| 13 | .606 | .430 | 2.33 | 0.97 | .51 | .42 | .58 |
| 14 | .424 | | 2.63 | 1.18 | .48 | | .66 |
| 15 | .603 | | 1.60 | 1.24 | .42 | | .40 |

*Note.* $λ_{Total}$ = factor loading in CFA of the sECS (one-factor model with correlated error terms among items addressing each emotion); $λ_{Mimicry}$ = factor loading in the CFA of the sECS-Mimicry (one-factor model); $rit_{Total}$ ($rit_{Mimicry}$) = part-whole corrected item-total correlation for the sECS (sECS-Mimicry); $P$ = response probability. Shapiro-Wilk Tests indicated deviations from normality for all items.

**Convergent and discriminant validity.** Both The sECS and the sECS-mimicry were significantly positively correlated with emotionality (*r* = .49 and *r* = .40, respectively), in principle arguing for the convergent validity of the scales, thought the relations were considerably higher compared to Study 1. Against expectations, the sECS scales showed small but significant ($p < .05$) positive associations with openness to experience ($r_{Total} = .09$, $r_{Short} = .12$). Consistent to expectations, no significant associations with honesty-humility (−.04, −.04), extraversion (−.08, −.05),and conscientiousness (−.02,.01) were found, arguing for discriminant validity, while the expected associations to agreeableness (.01, −.01)and the Geneva Emotion Recognition Test Short (.04,.02) could not be verified. The complete results (including confidence intervals and *p* values) can be found in Table 7. As displayed, the confidence intervals of the correlations to the Geneva Emotion Recognition Test Short are almost symmetrically placed around zero, demonstrating that the relations to emotion recognition could clearly not be verified.

**Longitudinal invariance.** Longitudinal measurement invariance was assessed by performing multiple-group CFA using lavaan [50]. In line with Mackinnon et al. [69], residuals from the same items were allowed to be correlated across waves (i.e., correlated error terms from Item 01 at the first measurement occasion and Item 01 at the second, third, and fourth occasions; the same for Item 02, etc.). This common practice accounts for the nonindependence of observations across waves [69]. For the present research, we assessed four types of invariance, namely, gross factor structure equivalence across measurement occasions (*configural invariance*), as well as three models that consecutively added equality constraints: the equivalence of factor loadings (*metric invariance*), intercepts (*scalar invariance*), and residual variances (*strict invariance*) across the occasions. In order to address the emotion-specific influences (as discussed above) without simultaneously inflating complexity, a one-factor model with correlated error terms among the items addressing each emotion was tested for the sECS. For the sECS-mimicry, a one-factor model was specified. In line with Mackinnon et al. [69], we constrained the variance of each latent variable to 1, instead of constraining the first factor loading. We computed CFI, RMSEA, SRMR, and TLI to evaluate the model fit. For comparing the models, we considered a cutpoint of >.01 indicating a substantial decrease in fit, while we deemed the differences in CFI (ΔCFI) the most relevant index, as it is the mostly established criterion for evaluating measurement invariance [70]. As shown in Tables 8 and 9, both the sECS and the sECS-mimicry demonstrated metric as well as partial scalar and strict invariance. Nevertheless, a limitation is the fit of the configural invariance model of the sECS, which was below established criteria for good model fit [53], especially regarding the SRMR. However, these results should be interpreted in light of the fact that the factor model was simultaneously modeled in *four* measurement occasions, wherefore a decrease in fit compared to the CFA reported

**Table 7. Correlations between the sECS/ sECS-mimicry and other measures of Study 2.**

| Scale | sECS | | | | | sECS-mimicry | | | | |
|---|---|---|---|---|---|---|---|---|---|---|
| | *r* | lb | ub | *p* | α (fdr) | *r* | lb | ub | *p* | α (fdr) |
| H | −.04 | −.14 | .05 | .818 | .043 | −.04 | −.13 | .05 | .792 | .043 |
| E | .49 | .42 | .56 | **<.001** | .007 | .40 | .32 | .48 | **<.001** | .007 |
| X | −.08 | −.17 | .02 | .943 | .05 | −.05 | −.14 | .04 | .864 | .05 |
| A | .01 | −.08 | .11 | .395 | .029 | −.01 | −.10 | .08 | .573 | .036 |
| C | −.02 | −.11 | .07 | .676 | .036 | .01 | −.08 | .10 | .42 | .029 |
| O | .09 | −.01 | .18 | .032 | .014 | .12 | .02 | .21 | .007 | .014 |
| GERT | .04 | −.09 | .17 | .273 | .021 | .02 | −.11 | .15 | .381 | .021 |

*Note.* Lb (ub) = lower (upper) border of the 95% confidence interval of the correlation coefficient. α (fdr) = Adjusted significance level according to the false discovery rate. H = Honesty-Humility, E = Emotionality, X = Extraversion, A = Agreeableness, C = Conscientiousness, O = Openness. GERT = Geneva Emotion Recognition Test Short. All *p* values are one-tailed. Significant *p* values in bold. Shapiro-Wilk Tests indicated deviations from normality for all items.

**Table 8. Series of invariance model comparisons for the sECS across measurement occasions (Satorra-Bentler corrected indices in parentheses).**

| Model | $\chi^2$ | df | p | RMSEA | CFI | Δ CFI | SRMR | TLI |
|---|---|---|---|---|---|---|---|---|
| Configural | 1550.933 (1470.233) | 1014 | <.001 | .063 (.058) | .865 (.860) | – | .115 (.115) | .850 (.844) |
| Metric | 1612.886 (1520.798) | 1050 | <.001 | .063 (.058) | .858 (.855) | .007 (.005) | .123 (.123) | .848 (.845) |
| Scalar | 1703.132 (1650.576) | 1086 | <.001 | .065 (.062) | .845 (.827) | .013 (.028) | .121 (.121) | .839 (.820) |
| *Scalar, rel. int. 05, 08, 03, 04* | 1646.504 (1570.785) | 1074 | <.001 | .063 (.059) | .856 (.847) | .002 (.008) | .121 (.121) | .849 (.840) |
| *Strict, rel. int. 05, 08, 03, 04* | 1701.023 (1602.398) | 1110 | <.001 | .063 (.058) | .851 (.849) | .005 (−.002) | .122 (.122) | .849 (.846) |

*Note. Re*l = relaxed; int. = equality constraint of intercepts. The following number(s) refer(s) to the item(s), in terms of which the respective equality constraint was relaxed.

**Table 9. Series of invariance model comparisons for the sECS-mimicry across measurement occasions (Satorra-Bentler corrected indices in parentheses).**

| Model | $\chi^2$ | df | p | RMSEA | CFI | Δ CFI | SRMR | TLI |
|---|---|---|---|---|---|---|---|---|
| Configural | 143.492 (142.286) | 94 | .001 | .063 (.062) | .955 (.947) | – | .081 (.081) | .943 (.933) |
| Metric | 161.510 (158.835) | 106 | <.001 | .063 (.061) | .950 (.942) | .005 (.005) | .097 (.097) | .943 (.935) |
| Scalar | 210.134 (215.655) | 118 | <.001 | .076 (.079) | .916 (.893) | .034 (.049) | .106 (.106) | .915 (.891) |
| *Scalar, rel. int. 05, 03* | 173.522 (172.772) | 112 | <.001 | .064 (.064) | .944 (.934) | .006 (.008) | .092 (.092) | .940 (.929) |
| *Strict, rel. int. 05, 03* | 180.399 (178.043) | 124 | .001 | .058 (.057) | .949 (.941) | −.005 (−.007) | .094 (.094) | .949 (.943) |

*Note. Re*l = relaxed; int. = equality constraint of intercepts. The following number(s) refer(s) to the item(s), in terms of which the respective equality constraint was relaxed.

above (which considered only the first measurement occasion), does not seem surprising. Anyway, the invariance results should be interpreted with caution.

**Temporal stability.** The temporal stabilities of the sECS and sECS-mimicry scales were examined on both the manifest level, by correlating sum scores for the items from each scale between measurement occasions, and the latent level (see below in parentheses). The latent stabilities were estimated by computing the correlations of the latent variables across occasions within the respective configural invariance model (see above). The sECS scores were strongly correlated across the intervals of 3–4 months, $r_{t1-t2}$ = .82 (latent correlation: .87), $r_{t2-t3}$ = .78 (.84), $r_{t3-t4}$ = .78 (.84); 6–8 months, $r_{t1-t3}$ = .75 (.83), $r_{t2-t4}$ = .79 (.85); and 9–12 months, $r_{t1-t4}$ = .81 (.85). The sECS-mimicry scores were also strongly correlated across the intervals of 3–4 months, $r_{t1-t2}$ = .73 (.90), $r_{t2-t3}$ = .76 (.93), $r_{t3-t4}$ = .70 (.79); 6–8 months, $r_{t1-t3}$ = .66 (.67), $r_{t2-t4}$ = .71 (.87); and 9–12 months, $r_{t1-t4}$ = .74 (.82). Taken together, Study 2 revealed high (sensu Cohen) [39] temporal stabilities for the sECS and sECS-mimicry across periods of up to 1 year.

## Study 3

With Study 3, we endeavored to replicate and extend the results of Studies 1 and 2 on the factor structure and the convergent and discriminant validity of the sECS and sECS-mimicry. On the basis of the conceptualization of SEC as a basic process underlying the broad construct of *affective empathy*, as well as previous research [28–30], we expected large positive associations between both sECS versions and affective empathy scales as well as moderate positive associations between both sECS versions and cognitive empathy scales (sensu Cohen) [39]. In particular, we predicted that both

sECS versions would be strongly positively related to the affective subscales of both the Basic Empathy Scale [37] and the Empathy Components Questionnaire [71] as well as the Toronto Empathy Questionnaire, which captures empathy as a primary affective phenomenon [16]. Moreover, we expected both sECS versions to be moderately positively associated with the cognitive subscales of the Basic Empathy Scale and the Empathy Components Questionnaire. In addition, we predicted a small (sensu Cohen) [39] positive association between both sECS versions and (a brief version of) the Reading the Mind in the Eyes Test [72], which measures emotion recognition [73].

Additionally, we expected a negative association between both sECS versions and narcissism: On the basis of the Trifurcated Model of Narcissism [74,75], narcissism can be modeled as a three-dimensional construct, with the factors *agentic extraversion*, *narcissistic neuroticism*, and *self-centered antagonism*. According to a recent meta-analysis [76], empathy appears to have a stronger negative relationship with self-centered antagonism than it does with the other two dimensions. The Narcissistic Admiration and Rivalry Questionnaire [77] consists of the two subscales *admiration* and *rivalry*, which are purported to measure agentic extraversion and self-centered antagonism, respectively [78]. Thus, we predicted that both sECS versions would show a small negative association with the Narcissistic Admiration and Rivalry Questionnaire subscale *admiration* and a moderate negative association with the Narcissistic Admiration and Rivalry Questionnaire subscale *rivalry* (sensu Cohen) [39].

Besides continuing the convergent and discriminant validity analyses, Study 3 also investigated the psychometric effects of rephrasing the content of item 03. As Study 2 revealed, item 03 showed a low factor loading and limited item-total correlation (especially within the sECS-mimicry). Even though one might solely attribute this to the positive valence of the item, an inspection of its content ("When someone smiles warm at me, I smile back and feel warm inside.") indicated another conceptual characteristic: This item obviously confounds two different stages of Hatfield's theoretical model of emotional contagion, namely *mimicry* ("smile back") and *contagion* ("feel warm inside"). Therefore, the item was reformulated by deleting the last phrase ("and feel warm inside"). We expected the item to show a slightly increased item-total correlation if administered in this reformulated version compared to the original wording for both the sECS and the sECS-mimicry. Likewise, we assumed the internal consistency of both scales to slightly improve by replacing the original item 03 by this reworded version.

## Method

**Participants and data collection.** The data were collected via an online survey (ww2.unipark.de) between April 08, 2024, and May 05, 2024. Participants were recruited by distributing flyers during lectures and via social media posts. Most participants were students at the university of Duisburg-Essen, Germany, who were given partial course credit or 5€ to compensate them for their participation. A total of 180 participants (76.1% women) aged between 18 and 64 years ($M = 23.18$, $SD = 6.57$; one missing value) completed all questionnaires (incomplete responses were deleted). Of the participants, 92.8% were students, 3.9% were employed, 1.1% were unemployed and 1.1% were retired. The study was approved by the Ethics Commission of the Department of Psychology of the University of Duisburg-Essen (No. EA-PSY25/23/25102023) and preregistered before data collection (https://osf.io/ncwyv). As in all studies, Participants provided written informed consent.

**Measures.** The internal consistencies and descriptive statistics of external measures can be found in Tables 11 and 12, respectively. The consistency and descriptive statistics of the sECS and sECS-mimicry can be found in the results section.

*sECS.* The German sECS was administered as described in Studies 1 and 2. A five-point response scale ($0 = never$, $1 = rarely$, $2 = sometimes$, $3 = often$, and $4 = always$) was used. In addition to the original wording of item 03, a reformulated version was administered, in which the last phrase was excluded, so that the reworded item reads: "When someone smiles warm at me, I smile back." To avoid biased responding as well as sequence effects, participants were randomly (but evenly) assigned to either answer the original version of the item at the beginning of the study and later in the study respond to the reformulated version – or *vice versa*.

*Basic Empathy Scale.* The German version of the Basic Empathy Scale was administered as described in Study 1.

*Empathy Components Questionnaire.* The Empathy Components Questionnaire [71] is a 27-item self-report instrument that addresses several subcomponents of both affective and cognitive empathy. These subcomponents are *drive* (i.e., the desire to emotionally engage with others) and *ability* (i.e., the skill to act accordingly) for both affective and cognitive empathy for a total of four subscales. Example items for the subscales *affective drive*, *affective ability*, *cognitive drive*, and *cognitive ability* are "I have a desire to help other people.", "I am good at responding to other people's feelings.", "I strive to see how it would feel to be in someone else's situation before criticizing them.", "I am usually successful in judging if someone says one thing but means another.", respectively. These four subscales are complemented by a fifth for assessing *affective reactivity* (i.e., the tendency to respond to and share the other person's emotions, example item: "I feel pity for people I see being bullied.") [71]. The German version as administered by Wieck et al. [79] was used. Responses were given on a five-point Likert scale (1 = *totally disagree*, 2, 3, 4, 5 = *totally agree*).

*Toronto Empathy Questionnaire*. The Toronto Empathy Questionnaire [16,41] was constructed by statistically forming a consensus between several established empathy questionnaires, resulting in a 16-item unidimensional scale. In order to address the field's problems of conceptual and operational heterogeneity [12], the Toronto Empathy Questionnaire was thereby designed to capture the empathy construct at the broadest range [16]. Responses were given on a five-point Likert scale (0 = *never*, 1 = *rarely*, 2 = *sometimes*, 3 = *often*, and 4 = *always*). An example item is "I find that I am "in tune" with other people's moods".

*Reading the Mind in the Eyes Test (Brief Version).* The Reading the Mind in the Eyes Test [72,80] assesses the ability to infer another person's mental state (emotions, thoughts) from a picture depicting the eyes of that person. For each of the 36 pictures (i.e., items), the participant is asked to select which of four terms best describes what the depicted person is thinking or feeling. Responses are coded 1 (*correct*) or 0 (*incorrect*). In this study, we used the 10-item brief version as introduced by Olderbak et al. [81].

*Narcissistic Admiration and Rivalry Questionnaire (brief version).* The Narcissistic Admiration and Rivalry Questionnaire [77] is an 18-item self-report instrument addressing two dimensions of grandiose narcissism, namely, *admiration* (i.e., seeking social admiration through self-promotion, example item: "I deserve to be seen as a great personality.") and *rivalry* (i.e., preventing social failure through self-defense, e.g., "Most people are somehow losers.") with three subscales each. In the present study, the six-item brief version [77,82] was administered, containing the two subscales *admiration* and *rivalry*. Responses were given on a six-point Likert scale (1 = *do not agree at all*, 2, 3, 4, 5, 6 = *agree completely*).

## Results and discussion

**CFA.** The CFA was conducted in line with Studies 1 and 2. As shown in Table 10, the one-factor model for the sECS showed weak fit to the data. The model with one factor and correlated error terms did not show good model fit either (instead of SRMR), but fitted the data significantly better than the one-factor model, $\Delta\chi^2(\Delta df = 12) = 126.651$, $p < .001$, whereas it was again outperformed by the bi-factor model, $\Delta\chi^2(\Delta df = 6) = 33.625$, $p = .008$. The latter showed acceptably incremental and excellently absolute fit to the data. Besides that, the one-factor model that was tested for the sECS-mimicry demonstrated excellent fit. Taken together, Study 3 replicated the results of Study 2 in terms of the dimensionality of both versions of the sECS, generally pointing out its factorial validity, while simultaneously indicating emotion specific influences on item responses, which we addressed via correlated error terms as well as a bi-factor model.

**Reliability.** The internal consistency values for the sECS were α = .75 (SE = .03) and ω = .81 (SE = .02), whereas the sECS-mimicry achieved values of α = .49 (SE = .06) and ω = .56 (SE = .10). Ordinal α and ω were also computed: The sECS achieved an ordinal α of .76 (SE = .03) and an ordinal ω of .76 (SE = .03). The sECS-mimicry reached an ordinal α of .49 (SE = .07) and an ordinal ω of .50 (SE = .06). IRT analyses were conducted parallel to study 2. Comparisons of the CCCs between the different response categories for every item (see S9 in S1 File) again indicated that the CCC of the middle category was in each case distributed between those from the two adjoint response categories.

**Table 10. CFA results for study 3 (Satorra-Bentler corrected test statistic in parentheses).**

| Scale | Model | χ² | df | CFI | TLI | RMSEA | SRMR | AIC |
|---|---|---|---|---|---|---|---|---|
| sECS | 1F | 218.272 (199.281) | 54 | .591 (.603) | .500 (.515) | .130 (.122) | .104 (.104) | 5756.175 |
| | 1F + CE | 91.621 (85.971) | 42 | .877 (.880) | .806 (.811) | .081 (.076) | .066 (.066) | 5653.523 |
| | BF | 57.996 (56.186) | 36 | .945 (.945) | .900 (.899) | .058 (.056) | .056 (.056) | 5631.898 |
| sECS-mimicry | 1F | 2.229 (2.153) | 2 | .994 (.995) | .981 (.986) | .025 (.021) | .029 (.029) | 1917.661 |

*Note*. 1F = One-factor model; 1F + CE = Model with one factor and correlated errors; BF = Bi-factor model. Shapiro-Wilk Tests indicated deviations from normality for all items.

These results generally replicate the findings of the first two studies. Besides that, participants' scores on the sECS ranged between 13 and 44, *M* = 28.9, *SD* = 6.1, while the sECS-mimicry scores ranged between 2 and 14, *M* = 8.6, *SD* = 2.4.

**Convergent and discriminant validity.** For the associations that the sECS and sECS-mimicry demonstrated with external measures, see Tables 11 and 12, respectively. The correlational pattern of the sECS generally matched our hypotheses. Deviations we found were the lack of correlations to the Narcissistic Admiration and Rivalry Questionnaire *admiration* subscale and the Reading the Mind in the Eyes Test. However, compared to all other measures, the Reading the Mind in the Eyes Test had a blatantly low internal consistency, questioning its interpretability. A low internal consistency of the Reading the Mind in the Eyes Test has been repeatedly reported before [81] and indicates the need for future research to further optimize this measure. Therefore, the results regarding this measure should be treated with caution. Moreover, we could indeed verify a negative correlation between the sECS and narcissistic rivalry, but the size was not very high – the upper boarder of the 95% confidence interval was −.03, calling the practical relevance of this finding into question. Apart from that, the correlation patterns of the sECS and sECS-mimicry resembled one another. For the sECS-mimicry, we could likewise confirm all correlations except for the Reading the Mind in the Eyes Test and (this time both subscales of) the Narcissistic Admiration and Rivalry Questionnaire. Taken together, these correlation patterns argue for the validity of both sECS scales.

**Table 11. Internal consistencies for external measures, correlations between the sECS and external measures of Study 3.**

| Measure – Scale | α (SE) | ω (SE) | Correlation to sECS | | | | |
|---|---|---|---|---|---|---|---|
| | | | r | lb | ub | p | α (fdr) |
| BES – AE | .67 (.04) | .81 (.03) | .69*** | .61 | .76 | **<.001** | .005 |
| BES – CE | .79 (.04) | .85 (.03) | .33*** | .19 | .46 | **<.001** | .023 |
| TEQ | .86 (.02) | .89 (.02) | .63*** | .53 | .71 | **<.001** | .009 |
| ECQ – CA | .58 (.06) | .73 (.04) | .23** | .09 | .37 | **.001** | .036 |
| ECQ – CD | .58 (.05) | .66 (.05) | .31*** | .17 | .44 | **<.001** | .032 |
| ECQ – AA | .66 (.05) | .75 (.04) | .32*** | .18 | .44 | **<.001** | .027 |
| ECQ – AD | .41 (.08) | .50 (.10) | .37*** | .23 | .49 | **<.001** | .018 |
| ECQ – AR | .64 (.04) | .74 (.04) | .49*** | .37 | .59 | **<.001** | .014 |
| NARQ-S – A | .80 (.03) | .80 (.03) | .03 | −.12 | .17 | .365 | .045 |
| NARQ-S – R | .64 (.06) | .65 (.06) | −.18** | −.32 | −.03 | .992 | .05 |
| RMET-S | .19 (.09) | .42 (.06) | .08 | −.07 | .22 | .145 | .041 |

Note. SE = Standard error. Lb (ub) = lower (upper) border of the 95% confidence interval of the correlation coefficient. α (fdr) = Adjusted significance level according to the false discovery rate. BES = Basic Empathy Scale; AE = affective empathy; CE = cognitive empathy. TEQ = Toronto Empathy Questionnaire; ECQ = Empathy Components Questionnaire; CA = cognitive ability; CD = cognitive drive; AA = affective ability; AD = affective drive; AR = affective reactivity; NARQ-S = Narcissistic Admiration and Rivalry Questionnaire Short Scale; A = admiration; R = rivalry; RMET-S = Reading the Mind in the Eyes Test Short Form. All p values are one-tailed. Significant p values in bold. Shapiro-Wilk Tests indicated deviations from normality for all items.

**Table 12. Descriptives of external measures and correlations between the sECS-mimicry and external measures of Study 3.**

| Measure – Scale | Descriptives | | | | Correlation to sECS-mimicry | | | | |
|---|---|---|---|---|---|---|---|---|---|
| | *M* | *SD* | *Min* | *Max* | *r* | lb | ub | *p* | α (fdr) |
| BES – AE | 28,9 | 6,1 | 13 | 44 | .58 | .48 | .67 | **<.001** | .005 |
| BES – CE | 8,6 | 2,4 | 2 | 14 | .26 | .45 | .65 | **<.001** | .009 |
| TEQ | 23,2 | 3,6 | 11 | 30 | .56 | .31 | .54 | **<.001** | .014 |
| ECQ – CA | 23,9 | 3,8 | 7 | 30 | .23 | .23 | .48 | **<.001** | .018 |
| ECQ – CD | 61,7 | 8,1 | 32 | 78 | .30 | .23 | .48 | **<.001** | .023 |
| ECQ – AA | 18,1 | 2,9 | 8 | 24 | .36 | .16 | .43 | **<.001** | .027 |
| ECQ – AD | 15,9 | 2,5 | 8 | 20 | .36 | .12 | .39 | **<.001** | .032 |
| ECQ – AR | 15,2 | 2,9 | 6 | 20 | .43 | .08 | .36 | **.001** | .036 |
| NARQ-S – A | 13,3 | 1,8 | 8 | 16 | .03 | −.12 | .18 | .342 | .041 |
| NARQ-S – R | 21,5 | 3 | 12 | 27 | −.12 | −.12 | .17 | .357 | .045 |
| RMET-S | 7,9 | 3,5 | 3 | 17 | .03 | −.26 | .03 | .939 | .05 |

*Note.* SE = Standard error. Lb (ub) = lower (upper) border of the 95% confidence interval of the correlation coefficient. α (fdr) = Adjusted significance level according to the false discovery rate. BES = Basic Empathy Scale; AE = affective empathy; CE = cognitive empathy. TEQ = Toronto Empathy Questionnaire; ECQ = Empathy Components Questionnaire; CA = cognitive ability; CD = cognitive drive; AA = affective ability; AD = affective drive; AR = affective reactivity; NARQ-S = Narcissistic Admiration and Rivalry Questionnaire Short Scale; A = admiration; R = rivalry; RMET-S = Reading the Mind in the Eyes Test Short Form. All *p* values are one-tailed. Significant *p* values in bold. Shapiro-Wilk Tests indicated deviations from normality for all items.

**Impact of item rewording.** The internal consistency values for the sECS including the reformulated item 03 instead of the original version were α = .74 (SE = .03) and ω = .80 (SE = .02), whereas the sECS-mimicry containing the reformulated item 03 instead of the original version achieved values of α = .45 (SE = .07) and ω = .52 (SE = .11). Ordinal α and ω were also computed: The reformulated sECS achieved an ordinal α of .76 (SE = .03) and an ordinal ω of .76 (SE = .03). The reformulated sECS-mimicry reached an ordinal α of .43 (SE = .08) and an ordinal ω of .46 (SE = .06). The internal consistency thereby did not increase after reformulating the item. Within the sECS, the part-whole corrected item-total correlation of item 03 decreased from an original $rit_{Total}$ = .46 to $rit_{Total}$ = .37 for the reworded version. For the sECS-mimicry, the part-whole corrected item-total correlation of the original item 03 was $rit_{Short}$ = .35, while the reworded item 03 yielded a $rit_{Short}$ = .27. Taken together, since neither the internal consistency, nor the item-total-correlation of item 03 benefited from the reformulation, this change was discarded for the final versions of both scales. In other words, both the final sECS and the final sECS-mimicry as analyzed and presented in the present research contain the original version of item 03.

## General discussion

In the current studies, we analyzed the psychometric quality of the German version of the Emotional Contagion Scale (sECS; from which the love items were excluded for theoretical reasons) and developed a mimicry brief version containing only the four *mimicry* items (sECS-mimicry). This mimicry brief version of the sECS is the most original contribution of the present research. In three studies, we examined the dimensionality and the convergent and discriminant validity of the scales in different samples (students in Studies 1 and 3; nursing staff in Study 2). We additionally assessed the internal consistency, psychometric properties, temporal stability, and longitudinal measurement invariance of the scales.

### Dimensionality

As explained in the Introduction, theoretical work on the construct of SEC [2] and the original study that introduced the ECS [20] suggested a unidimensional factor structure. Nevertheless, subsequent studies challenged the idea that the (simple) one-factor model was appropriate for the ECS (e.g., 25). To address this discrepancy, we tested two models in

addition to the one-factor model: A (rather parsimonious) one-factor model including correlated error terms among the items used to address each emotion as well as a (more complex) bi-factor model. Notwithstanding its complexity, the bi-factor model has the strongest a priori justification of the three models, both according to previous research pointing out to its superiority over other models [83] as well as in terms of theoretical consideration: Since susceptibility to emotional contagion (SEC) is assessed by the sECS via items tapping different discrete emotions, some items arguably share common error influences. These error influences can be modeled within CFA using first-level factors. By simultaneously estimating a bi-factor, one is possible to adhere to the original theory behind SEC that treats the construct as unidimensional [2,20]. In other words, the bi-factor model is a factor-analytic solution that seeks to integrate the theory behind SEC, which is originally unidimensional in nature, with psychometrically addressing the misfit of the simple one-factor model that is likely caused by emotion-specific error influences shared by certain groups of items.

As expected, the one-factor model showed weak fit for the sECS in all three studies. By contrast, the one-factor model with correlated error terms achieved considerably better fit according to AIC (and $\chi^2$) in all three studies, but only reached a fully acceptable model fit in Study 2. The moderate sample sizes might restrict the interpretability of the CFA in Studies 1 ($n = 195$) and 3 ($n = 180$) compared with Study 2 ($n = 442$). In addition, a limited response format (four-point scale) was used in Study 1, which is potentially accompanied with increased error variance since it forces participants to convert a neutral response into a high or low response [58]. Apart from that, a consistent finding across all three studies was the acceptable fit of the bi-factor model, that reached an excellent fit in Study 2 and consistently outperformed the other two models in all three studies according to AIC (and $\chi^2$).

On the one hand, CFA results from the present research replicated previous reports of the insufficiency of the one-factor model to describe the dimensionality of the ECS [25]. Notwithstanding, the present research also demonstrated that the move to (theoretically weakly justified) multidimensional solutions [25] is not necessary: As long as some shared error influences (expected a priori) on certain items are addressed in the CFA, a model entailing a general factor can achieve sufficient model fit. The phenomenon of ECS items sharing certain error influences can be explained by the item content, which always taps a certain emotion (e.g., Items 01, 04, and 14 are all related to the emotion sadness). As suggested by LoCoco et al. [28], each ECS item response is influenced by a general trait factor (SEC) as well as an emotion-specific factor.

These emotion-specific error influences can – in the most parsimonious way – be addressed by allowing the error terms of the respective items to be correlated with each other. However, as the results clearly showed, an even better fit could be achieved by a bi-factor model. The consistently good fit of the bi-factor model replicated Lo Coco et al.'s [28] results. One difference between the model with correlated errors and the bi-factor model is that the latter additionally allows the emotion-specific factors to be intercorrelated. The increase in fit in response to adding these model parameters (relationships between emotion-specific factors) was not surprising: Because experiencing a certain emotion (e.g., sadness) cannot be assumed to be independent from experiencing another emotion (e.g., happiness), allowing the factors that account for these emotion-specific influences to correlate (as the bi-factor model does) can be argued to explain additional variance in the data.

In three studies, we demonstrated the factorial validity of the German 12-item version of the ECS (sECS), which can be considered a unidimensional measure as long as emotion-specific influences on the item responses are simultaneously addressed in a CFA. Additionally, we developed and validated the sECS-mimicry. Whereas the Study 1 results were deemed rather pessimistic in terms of the fit of the one-factor model, Studies 2 and 3 found that this model offered an acceptable fit for the sECS-mimicry. In light of the shortcomings of Study 1 discussed above (moderate sample size, limited response format), these results overall suggest the factorial validity of the sECS-mimicry as well.

In contrast to the findings of the present study, a recently introduced questionnaire subdivides SEC into SEC for positive and SEC for negative emotions, demonstrating a clear two-dimensional structure according to the valence of emotions [22]. However, testing such a structure within the sECS does not appear very helpful, since it only includes a minority

of items with positive valence (three items tapping SEC for happiness). Nevertheless, Future studies could examine the intercorrelations between the unidimensional approach to SEC (as measured by the sECS) and this valence-based approach to further examine the validity of both scales. Besides that, future research could also explore the incremental predictive validity of both scales over each other in order to shed light on the question, how useful and practically relevant each approach can be considered.

## Convergent and discriminant validity

Study 1 demonstrated large positive correlations between both sECS versions and affective empathy scales and moderate positive associations with cognitive empathy scales. We replicated this pattern, which was also observed by previous studies [28,29], in Study 3 and it is consistent with the conceptualization of SEC as a basic affective empathic construct [18]. Considerably larger correlations to the rather affective subscales of the Interpersonal Reactivity Index compared to the rather cognitive scales were also reported for SEC for negative emotions, but not equally consistent for SEC for positive emotions [22], which is consistent with the idea that the sECS understands SEC as primary referring to negative emotions (as also reflected in its item contents). The pattern of results is also consistent with findings reported for the Questionnaire of Cognitive and Affective Empathy [84], whose subscale for "emotional contagion" was considerably closer related to the other affective-empathic subscales compared to the cognitive-empathic subscales [84,85].

By contrast, Studies 2 and 3 did not find significant associations between both sECS versions and behavioral-based measures of emotion recognition. The lack of correlations between different measurement approaches targeting empathic processes does not appear uncommon, as similar results have been previously reported for other self-report measures of empathy [12,86] as well as the ECS [87].

Additionally, and consistent with Wrobel and Lundqvist [29], the present research demonstrated small/moderate (Study 1) to strong (Study 2) positive associations between both sECS versions and neuroticism/emotionality. These relationships can be attributed to the emotional content of the sECS items and point out a certain confounding influence of neuroticism on responses to the sECS. Except for agreeableness (Study 1), no or weak associations with the other Big Five/HEXACO personality dimensions were found, arguing for the discriminant validity of the sECS and replicating the results of Wrobel and Lundqvist [29]. Moreover, Study 1 demonstrated a small positive association between the sECS and a self-report scale addressing altruism, whereas this association was not significant for the sECS-mimicry. These results are consistent with the idea that the often proposed link between empathy and altruistic or prosocial tendencies [38] more likely refers to higher order affective phenomena (empathic concern) than to the basic process of SEC. Consistent to our present findings, a positive association to helping behavior has also been reported for the Emotional Empathic Tendency Scale [23], which though refers to a total score that only partly refers to items measuring SEC. Finally, Study 3 demonstrated a negative association between the sECS and the Narcissistic Admiration and Rivalry Questionnaire subscale *rivalry*, whereas no association with the other subscale *admiration* was found, partially speaking for the convergent validity of the sECS.

Taken together, the correlational results emphasize the convergent and discriminant validity of both the sECS and the sECS-mimicry, which especially manifests in consistently high correlations to affective empathy, moderate correlations to cognitive empathy, and considerably smaller relations to theoretically less relevant constructs such as general personality (Big Five, HEXACO), with the exception of neuroticism, with which the measurement of SEC is arguably confounded.

## Longitudinal analyses

To our knowledge, the present study was the first to analyze the longitudinal measurement invariance of the (s)ECS as well as temporal stabilities across periods of up to 1 year. We found that both the sECS and the sECS-mimicry demonstrated metric invariance, as well as partial scalar and strict invariance, across the four measurement occasions (with 3–4 months between each assessment). However, a crucial limitation lies in the fit of the configural invariance model of the

sECS, which was below established criteria for acceptable model fit [53]. Nevertheless, the configural invariance model specified the underlying factor model simultaneously in *four* measurement occasions, arguably resulting in lower fit compared to a single occasion. Anyway, the invariance results should be interpreted with special caution. Future studies are needed to administer the (s)ECS in longitudinal settings and to compare the results between measurement occasions and to complement the herein presented invariance analyses.

Beyond that, the present research also revealed notably high temporal stabilities for the sECS in both versions (manifest 9–12-month stability for the sECS: $r_{tt} = .81$; sECS-mimicry: $r_{tt} = .74$). These results are in line with the conceptualization of SEC as a basal personality trait (*primitive empathy*) [2]. Thereby, these findings argue for the construct validity of the sECS and complement previous research on its test-retest reliability, which has been demonstrated for shorter intervals [20,25,29]. In addition, the high temporal stability of the sECS-mimicry appears especially remarkable considering that it comprises only four items and is consistent with the longstanding conceptualization of mimicry as a fundamental, unlearned behavior [88] (Lipps, 1906, as cited in Hatfield et al. [21], pp. 157–158). The fact that the internal consistencies were even lower for the sECS and sECS-mimicry than the temporal stabilities reflects the heterogeneity of the item content. Moreover, this finding is consistent with recommendations not to use internal consistency as a substitute for test-retest reliability in order to draw conclusions about a scale's validity [60].

## Limitations and future directions

Although our studies provide promising evidence for the validity, stability, and longitudinal invariance of both sECS and the sECS-mimicry across several samples, a limitation of the present research is its focus on the German context. Future studies are needed to replicate the suitability of the herein proposed mimicry brief version of the (s)ECS, which is the most original contribution of the present research, in other languages and cultural contexts. Moreover, the variance between the results of the three studies, that examined different samples (students/ nurses), are noteworthy, especially in terms of dimensionality. These differences could be explained by the low reliability of the sECS and the sECS-mimicry, which is a limitation of these measures.

Another limitation of the present research is that only Study 3 was preregistered, but not studies 1 and 2, which reanalyzed data that had been collected earlier. Furthermore, the present study computed manifest correlations for examining convergent and discriminant validity. Future studies could complement the findings of the present study by additionally employing structural equation modeling (SEM). Another limitation pertains to the number of correlations that were analyzed, bearing the risk of α error inflation, which accompanies multiple testing. However, we addressed this limitation by adjusting the significance level using the FDR method [62], which did not produce a substantially deviant pattern of results.

Apart from that, another important aspect of the present studies 1, 2, and 3 is that they were conducted during, before, and after the COVID-19 pandemic, respectively. These differences could influence each studies' findings, since there is evidence for a positive relation between SEC and concern about the spread of COVID-19 [89]. Therefore, the findings of Study 1, which was conducted during the pandemic, might be biased. On the other hand, Study 1 was conducted in 2022, i.e., after the time of the lockdowns in Germany. However, since there is also evidence showing that SEC moderates the relationship between COVID-19-related media consumption and elevated obsessive-compulsive symptoms [89], SEC definitely plays a role in the context of pandemic-related psychological strain and we should therefore compare the three studies' results with caution. This also seems recommendable in light of other global crises (e.g., wars in Ukraine and Israel) that arguably have been less globally salient in 2016 – the time in which Study 2 was conducted, additionally complicating comparisons between the three studies. Indeed, the descriptive statistics do not indicate a substantial difference in the mean emotional contagion between Study 2 (2016, M = 27.6, SD = 6.1) and 3 (2024, M = 28.9, SD = 6.1). However, in light of the different samples (nurses vs. students), a comparison remains difficult.

Apart from that, the present research focused on highly educated and predominantly female participants. Especially the large share of women emphasize that the findings should be treated with caution and not be simply generalized to all genders. Future studies are needed that examine the sensitivity of findings (e.g., factor structure of the sECS, invariance, correlations to other measures) for the influence of participants' gender. Moreover, the present research focused on student samples (Studies 1 and 3), from which most were psychology students and on nurses (Study 2). Both professions involve the frequent contact with people and caring about other peoples' well-being, which arguably might limit the variance in studied constructs, such as empathy and SEC. Finally, a fundamental shortcoming of the (s)ECS *per se* is the reliance on self-report, which requires participants to consciously reflect on a process conceptualized as automatic (e.g., 21). Of course, self-report is a low-threshold and parsimonious measurement approach, paving the way for many more studies and larger samples. Nevertheless, future research needs to complement the (s)ECS literature by validating behavioral-based approaches to mimicry and emotional contagion.

### Implications and practical applications of the present research

The present research presents as well a validated German version of the (susceptibility to) Emotional Contagion Scale (sECS), as a newly developed mimicry brief version (sECS-mimicry). Both scales can be used as well for research purposes as in various practical contexts, e.g., individual assessment in the clinical or work context. For instance, the relevance of susceptibility to emotional contagion and mimicry in the clinical context is underscored by previous reports of affective empathy deficits in depressive patients [90], suggesting that depressive mood suppresses the process of emotionally resonating with others. On the contrary, there is also evidence for positive associations between empathy and psychological strain [91] and the phenomenon of depressive mood spreading across humans, which has been labeled *contagious depression* [92]. These results emphasize the complexity of the relationship between SEC and psychological health, also stressing the importance of valid measurement tools for future research to further investigate this matter.

Particularly, the sECS mimicry seems especially suitable for projects that focus on the overt mimicry subconstruct of susceptibility to emotional contagion as well as cases of very limited resources, such as large surveys with many measures.

Since the bi-factor model achieved good fit in the present study, a unidimensional scoring of the sECS seems acceptable, instead of computing subscales for each discrete emotion category [29]. As explained, the present findings are compatible with the theory of emotional contagion as a unidimensional construct [2], but further investigation in terms of its dimensionality and potential differences for positive and negative emotions seems reasonable [22]. The remarkably high longitudinal associations of both the sECS and the sECS-mimicry also point out to the general relevance of the constructs of susceptibility to emotional contagion and mimicry within personality psychology – a field that per definition examines temporally stable traits [93] and thus unsurprisingly pays special attention to longitudinal associations (as typically assessed via test-retest correlations in shorter intervals) [60].

### Supporting information

**S1 File.   Study 1 CFA results with the DWLS estimation method. S2. Results for an ECS version including items 06, 09, and 12. S3. Data Study 1. S4. Data Study 2. S5. Data Study 3. S6. ECS items.S7. Power analyses.S8. CCCs study 2. S9. CCCs study 3.**
(ZIP)

### Author contributions

**Conceptualization:** Tobias Janelt, Tobias Altmann, Marcus Roth.

**Formal analysis:** Tobias Janelt, Danièle Anne Gubler.

**Funding acquisition:** Tobias Altmann, Marcus Roth.

**Methodology:** Tobias Janelt, Tobias Altmann, Danièle Anne Gubler.

**Project administration:** Marcus Roth.

**Supervision:** Tobias Altmann, Marcus Roth.

**Writing – original draft:** Tobias Janelt.

**Writing – review & editing:** Tobias Janelt, Tobias Altmann, Danièle Anne Gubler, Marcus Roth.

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
