## [Decision Letter · Decision Letter 0]

24 Jun 2025

Dear Dr. Janelt,

Thank you for submitting your manuscript to PLOS ONE. After careful consideration, we feel that it has merit but does not fully meet PLOS ONE’s publication criteria as it currently stands. Therefore, we invite you to submit a revised version of the manuscript that addresses the points raised during the review process.

Thank you for your manuscript. I have invited three very professional reviewers, which have indicated several areas for revision. Please carefully address these, and I will be happy to consider your revised paper for publication.

We look forward to receiving your revised manuscript.

Kind regards,

Paweł Larionow, Ph.D.

Academic Editor

PLOS ONE

2. We note that there is identifying data in the Supporting Information file < Supporting Information.zip>. Due to the inclusion of these potentially identifying data, we have removed this file from your file inventory. Prior to sharing human research participant data, authors should consult with an ethics committee to ensure data are shared in accordance with participant consent and all applicable local laws.

-Location data

Please remove or anonymize all personal information, ensure that the data shared are in accordance with participant consent, and re-upload a fully anonymized data set. Please note that spreadsheet columns with personal information must be removed and not hidden as all hidden columns will appear in the published file.

Reviewers' comments:

Reviewer's Responses to Questions

**Comments to the Author**

1. Is the manuscript technically sound, and do the data support the conclusions?

Reviewer #1: Yes

Reviewer #2: Yes

Reviewer #3: Partly

2. Has the statistical analysis been performed appropriately and rigorously?

Reviewer #1: Yes

Reviewer #2: Yes

Reviewer #3: Yes

3. Have the authors made all data underlying the findings in their manuscript fully available?

Reviewer #1: Yes

Reviewer #2: Yes

Reviewer #3: Yes

4. Is the manuscript presented in an intelligible fashion and written in standard English?

Reviewer #1: Yes

Reviewer #2: Yes

Reviewer #3: Yes

Reviewer #1: I would like to thank you for inviting me to review this manuscript. I have read the article carefully. While it covers an interesting topic, some concerns lead me to recommend major revisions. My reasons for this decision are as follows:

ABSTRACT

The title mentions the nomological network, but the abstract does not explicitly address it. Including a brief explanation of how the ECS fits within the broader theoretical framework or its relationships with other constructs would enhance the depth of analysis.

While factor models are mentioned, a brief explanation of the dimensionality and what specific dimensions are being assessed could provide clearer insights into the scale's structure.

Adding a sentence about the practical implications or applications of the ECS-Total and ECS-Short could help readers understand the significance of the research beyond its theoretical contributions.

More details on the selection process and rationale for the items included in the ECS-Short could be beneficial, especially considering its development is a key aspect of the study.

INTRODUCTION

While the introduction provides a solid foundation, it could benefit from a deeper exploration of prior studies concerning the ECS's psychometric properties in different cultural contexts to strengthen the rationale for the current study.

The introduction could clarify the distinction between ECt and other components of empathy more explicitly, potentially by providing more examples or discussing the implications of these distinctions in greater detail.

The introduction briefly mentions the nomological network but does not delve into what this entails specifically in the context of emotional contagion. Expanding on this concept could provide a clearer understanding of its relevance and importance.

Including references to relevant interdisciplinary research (e.g., from neuroscience or sociology) could deepen the analysis and demonstrate the broader implications of studying emotional contagion.

DISCUSSION

While the discussion provides a good analysis of the ECS, it could benefit from a more detailed comparison with other emotional contagion measures or related scales to highlight its unique contributions or advantages.

The discussion could be enhanced by elaborating on the broader implications of the findings for the field of psychology, particularly in understanding emotional contagion and its measurement.

Including potential practical applications of the ECS and ECS-Short in various settings (e.g., clinical, organizational) would extend the discussion's relevance.

While the analysis of dimensionality is thorough, further clarification on the theoretical justification for choosing the bifactor model over others could strengthen the argument.

The discussion could integrate more recent studies or theories to support or contrast the findings, providing a more comprehensive literature backdrop.

Reviewer #2: Overall. Thank you for this very fascinating series of emotional contagion scale (ECS) studies, which may offer significant contributions both theoretically (factor structure and short form) and practically (a German adaptation). Most notably, the bi-factor model fit and mimicry item short form are theoretically compelling. Also, confirmation of emotional contagion as a 'trait' adds to the theoretical database. Practically, having documented a German language adaptation may be of practical value to clinical and research practitioners. Below I offer some thoughts about your research and manuscript, which you may consider for revision.

Title.

The full title is excessively descriptive and yet inadequately describes your research. Your research is very extensive, so describing it in a title seems challenging. Given the theoretical value of your studies, not to mention the German adaptation, I recommend using the short title emphasizing 'validation', a 'German' version, and 'a brief form'. Also, the full title refers to 'nomological network', which is I think may get lost in the manuscript as you do not identify any of your studies as such.

Keywords.

The keywords are ideal for identification of the studies.

Abstract.

The abstract provides research details (theoretical purpose, design, findings), but presumes some valuable implication. Though limited with respect to words, a comment about theoretical and practical value may enhance the abstract, even if it requires abbreviating other parts.

Introduction.

Overall. The theoretical framework is clearly provided. Essential constructs and variables are identified along with the purpose(s) of the current research.

1. Comment. When introducing the current research, the section 'The Present Studies' states the purpose is ". . . aimed at validating the German version of the ECS, including its factor structure, nomological network, longitudinal measurement invariance, and temporal stability." Developing and validating a brief version is described as a second purpose. In your introductory paragraph you reference and describe 'three studies'. You then proceed with presentation of each study 1, 2, and 3, each complete with statement of purpose, Method, Measures, Results, and Discussion.

2. Suggestion. The presentation of these studies as described above is confusing. There are 3 studies, each essentially valuable theoretically, yet the actual purpose of each is not clear until the reader has completed reading the entire series of studies. I suggest reconsider presentation of the studies and corresponding results.

(a). Prior to presenting the actual studies, in the general introduction to the studies (The Present Studies), make clear briefly what each of the three study purposes are. Maybe have each study be subheaded with the purpose and design/methods. Then, have a Results section, with subheaded studies, for each a brief restatement of purpose followed by the results. Then, have a Discussion section, again, each study subheaded with corresponding Discussion.

(b). One confusion pertains to your summary tables, e.g., Table 1, Table 2, each of which have the results for the different Studies, though they are presented within Study 1.

My suggestion is basically a reorganization of studies such that the studies 1, 2,3 are nested with (a) purposes and methods, (b) Results, and (c) Discussion.

Methods.

Overall. Your methods are generally complete with ample description of the models and modeling procedures.

1. Recommendation. While you may be applying generally accepted procedures used in cited research, consider offering a citation of standards for language adaptation of clinical measures and corresponding research designs, e.g., International Test Commission (ITC) and standards for measurement language adaptation. https://www.intestcom.org/.

2. Recommendation. Study 1 forms used a 4-point Likert response scale, Study 2 forms expanded the response scale to a 5-point Likert. This is a nontrivial change to the forms, one which may go beyond simply obtaining higher internal consistency. At least, some discussion may be needed to explain the process underlying this finding. Furthermore, changing the response format may in fact change the measured construct. Fitting an item response model to the data to understand the response process would be very valuable, particularly for the brief form of mimic items. Note, I am not necessarily suggesting this for this set of studies as a revision, but without question, more attention is called for.

3. Comment. For the convergent and discriminant validity studies, beyond a visual inspection of the measurement correlation matrices with respect to the hypothesized covariances, a more rigorous structural equation model should be fit for hypothesis testing. On the other hand, given the obtained data, fortunately for the researchers, the 'interocular traumatic test' suffices, i.e., you know it when it hits you between the eyes. Again, however, an SEM model could have been fitted to the matrices to test the hypotheses. One of countless relevant citations is, Raykov, Tenko (2011). Evaluation of convergent and discriminant validity with multitrait-multimethod correlations. British Journal of Mathematical and Statistical Psychology, 64, 38-52.

4. Recommendation. Along with validity results, I recommend always providing point reliability estimates with associated standard errors. These would be estimated as parameters using structural equation modeling procedures.

Results.

Overall. Results from analyses per study are well described. As noted above, the presentation is sometimes confusing and required several readings to fully appreciate the purpose and results of the series of studies. The abbreviated 'brief' form is most interesting and from some perspective may constitute the most fascinating and certainly original contribution.

1. Suggestion. Table the results for each study independently as described above.

Discussion.

Overall. There are many findings to be discussed. You return to your research purpose with interpretation of your findings.

1. Comment. The long title mentions nomological network, and while the validity studies may be intended to complete this inquiry, the findings are not discussed as such, i.e., nomological network. This would constitute an important theoretical contribution and findings should be conceptualized in this term.

2. Recommendation. Highlight the brief form as an original contribution.

3. Comment. The implications of psychometric research are sometimes treated as 'needless-to-say'. In this case, you have much to say about implications and should include them.

Writing.

Overall. This is a well written manuscript, and was a pleasure to read. Thank you.

Reviewer #3: Review

Journal name: PLOS ONE

Manuscript number: PONE-D-25-13429

Manuscript title: Assessing emotional contagion: Dimensionality, nomological network, longitudinal invariance, and 1-year stability of the German emotional contagion scale and development of a brief version

Date received: May 22, 2025

Date of review: June 23, 2025

Suggestion: Major revision

Comments to the Authors

Thank you for the opportunity to review this manuscript!

The manuscript seems promising and especially the development of the presented brief scale could be interesting for future studies.

However, it is a very complex paper with a very large number of reported findings and potential limitations. Thus, it seems necessary that the writing and reporting is particularly consistent, transparent, and understandable!

Overall, I see a number of problems that could be addressed in a major revision.

Title:

Emotional contagion represents an interpersonal process between two or more individuals. The ECS aims at assessing in individual's susceptiblity to emotional contagion. Hence, "assessing emotional contagion" is misleading and incorrect.

Generally, the name of the ECS scale was misleading from the beginning (it does not measure the process emotional contagion, even if Doherty says so). Further, the reported scale does not contain all items of the ECS. For these reasons, I am not convinced that the reported scale should adapt the initially misleading name ECS. The short version, thus, does not aptly represent a short version of the original ECS, but instead rather seems to represent a "mimicry version" of your modified ECS (German version) which could be an interesting addition for future studies.

Abstract:

To me it seems that a lot of terms are brought up here that are used ambiguously in different areas of research (mimicry, synchronization). For the purpose of clarity, it seems helpful to stick to the central topic of the manuscript (susceptibility to emotional contagion) and prevent the misconception of assessing emotional contagion with the ECS.

It seems unusual to use abbreviations in the abstract.

To avoid misunderstandings, better report sample sizes of the three studies separately.

The findings on the models' fit are misleadingly reported. The fit indices did not support "satisfactory fit" for both the 1F+CE and the BF model.

Introduction:

Again, emotional contagion represents an interpersonal process between two or more individuals. This process should be clearly delineated from an individual's susceptiblity to emotional contagion which seems to be what you aim to investigate in this paper. Please review your use of emotional contagion throughout the manuscript and be more precise about the concept you are investigating.

Further, the conceptual ambiguity of the concept empathy should be discussed and both emotional contagion and susceptibility to emotional contagion should be delineated from empathy and/or different empathy-subfacets/concepts.

You are almost exclusively citing Hatfield and colleagues. Other references should be included on emotional contagion and susceptibility to emotional contagion.

Mimicry has been conceptualized very differently in psychological research regarding socio-emotional processes (eg Hess or Dimberg etc) which should be discussed alongside Hatfield's use of the term.

Explain/justify why you use a model from the context of psychopathy and not a more general model given the non-clinical nature of your samples.

Throughout the introduction, many terms are brought up but not defined, eg empathic concern, emotion awareness. This makes it hard to follow for the reader. Consider briefly defining these concepts.

The selection of discrete emotions in the ECS should be discussed more critically. They do not represent any commonly reported model. Discuss general critique regarding discrete emotion models. Discuss the ambiguity and overlap of some of the items, eg fear with stress.

Other self-report measures of an individual's susceptibility to emotional contagion should be discussed, eg emotion-specific or positively-negatively valenced scales.

The use of a very large number of abbreviations makes it hard to read and follow, eg EC, PT, FT, IRI, etc. Consider dropping the abbreviations and writing the full words instead.

Explain/justify why you still consider your scale(s) a version of the ECS after excluding 3 items only based on your theoretical (but very reasonable) critique on the love items?

Can your scale really be called an ECS scale then?

Explain why you kept the rest of the discrete emotions items?

The ECS short rather seems to be a subscale entailing mimicry items and not just a short version. Overall, the idea of a short ECS-mimicry scale is interesting, eg to assess a more salient and observable aspect of the complex process of ec, but this scale could, thus, be more aptly called some sort of mimicry-version of the modified ECS (without the love items), eg susceptibility to mimic emotional expressions.

Study 1:

Explain why the TEQ was not used in study 1.

Alpha = .05 seems to be too high given the number of correlations due to the problem of multiple testing and alpha inflation.

Clarify whether power analyses are in the supplement, if not please add them.

Add example items to the descriptions of the measures (in all studies and for all measures).

Be consistent in using either joy or happiness (eg page 11, line 233).

Why did you use only one subscale of the FAB scale?

For model comparison, the AIC should be used. The analyses, results, and discussion should be revised accordingly.

In general, be more detailed and explicit about the standards for fit indices you are applying. Specify the cutoffs for the fit indices more precisely (with references).

Discuss the model fit parameter of the models 1F and 1F+CE more thoroughly and more critically.

TLI for the BF model is below the cutoff for a good model fit (.95) according to Hu & Bentler (1999), this should be stated and discussed more clearly!

The fit of the ECS short is not acceptable! this should be stated and discussed more clearly.

Alpha and omega for both scales are low, this should be discussed. Provide references for the claim that a response scale comprising 4 answers can/should be considered ordinal.

Correlations should be summarized in the text, "largely in line" is not sufficient here.

The table's format is crooked and out of line.

What is the last column?

In the sample description you say 192 individuals filled out the survey, here you report up to 301 for study 1? Which number is correct?

The title of the table says study 1 and 3, but the column "study" says 1 and 2.

Were the measures normally distributed?

Please report more detailed descriptive statistics for all measures in all studies.

I am having a hard time with the not-preregistered "predictions". Explain and justify why the analysis of study q (and study 2 ) apparently were not preregistered?

How do you tackle the uncertainty that arises from the fact that these analyses were not preregistered?

This needs to be discussed.

Study 2:

Explain more precisely why nursing staff was chosen, eg also very high share of women?

Explain/justify why data from 2016 is used.

Was the 2016 study also approved by the ethics committee?

Clarify the intervention conducted in the overarching project.

Add sample information for sample 2 and 3 (eg gender, age, etc) and consider using a table for all three samples.

Clarify (for all studies and all measures), whether only the answers 0 and 4 (or 5) were presented to the participants with words or all answers (i.e., 0, 1, 2, 3, 4, 5). Please provide all answer options that were presented.

Again, for model comparison, the AIC should be used. The analyses, results, and discussion should be revised accordingly.

Again, TLI is below the cutoff for a good model fit (.95) according to Hu & Bentler (1999), this should be stated and discussed more clearly!

Discuss the TLI value > 1 for the ECS short in study 2.

Was ordinal alpha and omega calculated here?

Explain why you consider 4 response options ordinal and 5 continuous (enough)?

Summarize the descriptive statistics and psychometric properties in the text and report them in a table instead of the supplement.

Discuss the model fit of the configural invariance model more thoroughly (RMSEA and CFI are not good) and also report SRMR and TLI.

Justify your choice of delta CFI as index for invariance (why not delta chi square or delta RMSEA)?

Be consistent in your use of the terms, eg residual or strict invariance, in the text and tables.

Be consistent in how you report "rel. int. 05, ...", either in parenthesis or after a comma.

Justify why it seems more acceptable to you to modify the original response scale (4 options) than to keep it, given that internal consistency only "increased slightly".

Study 3:

Does the ethics approval only pertain to study 3? Were study 1 and 2 also approved by the ethics committee? Please clarify.

Explain why the preregistration was registered on april 8 and the data collection started in april 4 2024, but it says "preregistered before data collection" in the text.

Discuss the large share of women in the sample (in all three studies). Consider conducting sensitivity analyses for all studies regarding participants' gender.

Clarify whether all response options were presented to the participants with words or only the extremes (for all measures).

Add example items for all measures in all studies.

Again TLI below .95 is not excellent.

Report the AIC and compare the models accordingly (for all studies and model comparisons).

In the preregistration it says that one item was reformulated and tested. I cant find anything on that analysis step in the manuscript. Please clarify.

The potential influence of low alpha values (eg as low as .19) on the correlations should be discussed.

Better report and discuss confidence intervals for all correlations in all studies.

Overall, consider summarizing results and discussion in a combined "Results & Discussion" section (for all studies).

General Discussion:

The dimensionality section of the discussion needs to be revised:

The model fit of the 1F+CE model was not "close to the threshold for a good fit in studies 1 and 3" and the TLI was not excellent either in study 2.

The model fit of the BF model was only excellent in study 2, but not in study 1 and 3 (TLI < .95).

The findings should be discussed accordingly.

The model comparisons should be based on the AIC. The results should be reported transparently and discussed accordingly.

In the convergent and discriminant validity section of the discussion, a concluding remark is missing regarding your evaluation of the scales' validity.

The longitudinal analyses section of the discussion might need to be revised depending on the revised findings, especially regarding the missing model fit indices.

Emotional contagion is not a personality trait (line 631). It is an interpersonal process. What you are investigating is an individual's susceptibility to emotional contagion.

In line 632, "Hatfield et al." seems to be wrong.

The limitations section of the discussion is very brief:

Discuss the risk of alpha inflation due to multiple testing for all studies more thoroughly.

Discuss the samples' homogeneity in all three studies more thoroughly (women, profession, etc).

Discuss the different time points of the three studies with respect to the COVID-19 pandemic more thoroughly (before, during, and after the pandemic) and discuss the potential impact of the time point of study 2 (2016) being 6/8 years before the time points of studies 1 and 3.

Discuss the lack of preregistrated analyses in study 1 and 2.

Include practical implications of your research in the discussion.

Language and Style:

Some commas are missing.

Sometimes blank spaces are missing between parentheses.

APA level 2 headings are written in title case.

Paragraphs should be indented consistently.

Sentences should not start with numbers.

**Do you want your identity to be public for this peer review?** For information about this choice, including consent withdrawal, please see our Privacy Policy

Reviewer #1: No

Reviewer #2: **Yes: ** Paul Yovanoff

Reviewer #3: **Yes: ** Anton Marx

---

## [Author Response · Author response to Decision Letter 1]

24 Jul 2025

[Copy of the "Response to Reviewers" file]

Dear Dr. Larionow,

Thank you very much for taking our manuscript into consideration and the invitation to resubmit a revision. We would especially like to thank you and the reviewers for the valuable feedback and comments. We believe the revised manuscript to be substantially improved. We explicitly addressed each point in the table below.

Yours sincerely,

The Authors

Editor‘s / Reviewer’s suggestions Response

Journal requirements

01 Please ensure that your manuscript meets PLOS ONE's style requirements, including those for file naming. We ensured to meet PLOS ONE's style requirements and also checked the file names.

02 We note that there is identifying data in the Supporting Information file < Supporting Information.zip>. Due to the inclusion of these potentially identifying data, we have removed this file from your file inventory. Prior to sharing human research participant data, authors should consult with an ethics committee to ensure data are shared in accordance with participant consent and all applicable local laws. Thank you for highlighting this point. We have now removed all demographic variables from the data frames (S4, S5, S6) and uploaded the <supporting information.zip> again, which should thus now be fully anonymized.

Reviewer #1:

I would like to thank you for inviting me to review this manuscript. I have read the article carefully. While it covers an interesting topic, some concerns lead me to recommend major revisions. My reasons for this decision are as follows: Thank you very much for taking the time to review our manuscript.

03 The title mentions the nomological network, but the abstract does not explicitly address it. Including a brief explanation of how the ECS fits within the broader theoretical framework or its relationships with other constructs would enhance the depth of analysis. We understand that we used the term “nomological network” a bit too careless. Indeed, we wanted to refer to the concept of convergent validity, which is examined in terms of correlations to external measures. We have revised the wording in the title and the whole manuscript to avoid any misunderstandings associated with using the term “nomological network”.

04 While factor models are mentioned, a brief explanation of the dimensionality and what specific dimensions are being assessed could provide clearer insights into the scale's structure. We added a sentence in the abstract, clarifying that the mentioned fit of the factor models suggests factorial validity, since the ECS is aimed to measure a general factor.

05 Adding a sentence about the practical implications or applications of the ECS-Total and ECS-Short could help readers understand the significance of the research beyond its theoretical contributions. Thank you for highlighting this point. We added a sentence in the abstract, emphasizing the theoretical implications and practical applications of the present research.

06 More details on the selection process and rationale for the items included in the ECS-Short could be beneficial, especially considering its development is a key aspect of the study. For the ECS-Short we selected the four items of the total questionnaire that explicitly address the process of mimicry. We revised the wording in the abstract to point this out more clearly.

07 While the introduction provides a solid foundation, it could benefit from a deeper exploration of prior studies concerning the ECS's psychometric properties in different cultural contexts to strengthen the rationale for the current study. Thank you for this suggestion. We added details about prior validation studies’ findings on the dimensionality and convergent validity of the ECS in order to provide the reader with a clearer and more comprehensive background of the present research.

08 The introduction could clarify the distinction between ECt and other components of empathy more explicitly, potentially by providing more examples or discussing the implications of these distinctions in greater detail. Thanks for this suggestion as well. We revised the subsection that delineates empathy and emotional contagion, now providing deeper insights into the conceptual differences and implications.

09 The introduction briefly mentions the nomological network but does not delve into what this entails specifically in the context of emotional contagion. Expanding on this concept could provide a clearer understanding of its relevance and importance. As stated above, we understand that we may have been using the term “nomological network” too loosely. We thus revised our wording in the manuscript as well as the title to refer to the concept of convergent and discriminant validity, which is examined via the correlations to external measures.

10 Including references to relevant interdisciplinary research (e.g., from neuroscience or sociology) could deepen the analysis and demonstrate the broader implications of studying emotional contagion. We have added references to relevant interdisciplinary research, e.g., neuroscience, biological / physiological psychology, political psychology to provide the reader with a more comprehensive picture of the broad relevance of emotional contagion.

11 While the discussion provides a good analysis of the ECS, it could benefit from a more detailed comparison with other emotional contagion measures or related scales to highlight its unique contributions or advantages. Thank you for your suggestion. We added comparisons to other self-report measures of SEC including references in the general discussion. Please note that there are not many other such questionnaires available and except one, they merely address our construct of interest with one subscale among others, but we anyway connected our findings to the findings reported for these questionnaires.

12 The discussion could be enhanced by elaborating on the broader implications of the findings for the field of psychology, particularly in understanding emotional contagion and its measurement. We complemented the general discussion by a last section, pointing out to the implications of the present findings for the field of psychology, as well as the practical applications of the scales examined.

13 Including potential practical applications of the ECS and ECS-Short in various settings (e.g., clinical, organizational) would extend the discussion's relevance. We added potential practical applications at the end of the general discussion.

14 While the analysis of dimensionality is thorough, further clarification on the theoretical justification for choosing the bifactor model over others could strengthen the argument. Thank you for this suggestion. We added a more detailed explanation for the theoretical justification of the Bi-Factor model over other models in the dimensionality section of the general discussion.

15 The discussion could integrate more recent studies or theories to support or contrast the findings, providing a more comprehensive literature backdrop. We complemented the discussion by a more comprehensive comparison with relevant previous studies.

Reviewer #2

Overall. Thank you for this very fascinating series of emotional contagion scale (ECS) studies, which may offer significant contributions both theoretically (factor structure and short form) and practically (a German adaptation). Most notably, the bi-factor model fit and mimicry item short form are theoretically compelling. Also, confirmation of emotional contagion as a 'trait' adds to the theoretical database. Practically, having documented a German language adaptation may be of practical value to clinical and research practitioners. Below I offer some thoughts about your research and manuscript, which you may consider for revision. Thank you very much for your time and the appreciation you expressed.

16 The full title is excessively descriptive and yet inadequately describes your research. Your research is very extensive, so describing it in a title seems challenging. Given the theoretical value of your studies, not to mention the German adaptation, I recommend using the short title emphasizing 'validation', a 'German' version, and 'a brief form'. Also, the full title refers to 'nomological network', which is I think may get lost in the manuscript as you do not identify any of your studies as such. Thank you for highlighting this point. We have shortened the title, while trying to keep the most important information. We also omitted the term “nomological network”, which understandably may have been misleading at this point.

17 The abstract provides research details (theoretical purpose, design, findings), but presumes some valuable implication. Though limited with respect to words, a comment about theoretical and practical value may enhance the abstract, even if it requires abbreviating other parts. Thank you for your suggestion. We added a sentence in the abstract, emphasizing the theoretical implications and practical applications of the present research.

18 Overall. The theoretical framework is clearly provided. Essential constructs and variables are identified along with the purpose(s) of the current research.

1. Comment. When introducing the current research, the section 'The Present Studies' states the purpose is ". . . aimed at validating the German version of the ECS, including its factor structure, nomological network, longitudinal measurement invariance, and temporal stability." Developing and validating a brief version is described as a second purpose. In your introductory paragraph you reference and describe 'three studies'. You then proceed with presentation of each study 1, 2, and 3, each complete with statement of purpose, Method, Measures, Results, and Discussion.

2. Suggestion. The presentation of these studies as described above is confusing. There are 3 studies, each essentially valuable theoretically, yet the actual purpose of each is not clear until the reader has completed reading the entire series of studies. I suggest reconsider presentation of the studies and corresponding results.

(a). Prior to presenting the actual studies, in the general introduction to the studies (The Present Studies), make clear briefly what each of the three study purposes are. Maybe have each study be subheaded with the purpose and design/methods. Then, have a Results section, with subheaded studies, for each a brief restatement of purpose followed by the results. Then, have a Discussion section, again, each study subheaded with corresponding Discussion.

(b). One confusion pertains to your summary tables, e.g., Table 1, Table 2, each of which have the results for the different Studies, though they are presented within Study 1.

My suggestion is basically a reorganization of studies such that the studies 1, 2,3 are nested with (a) purposes and methods, (b) Results, and (c) Discussion. Thank you for the appreciation you expressed and for your suggestions. We added a specific description of the three studies’ purposes in the general introduction (subsection “The present studies”). We first tried to implement your suggestion to reorganize the presentation of the three studies as nested within (a) purposes and methods, (b) Results, and (c) Discussion. However, this confronted us with another issue: In our present case, the studies were not conducted parallel to each other, but consecutively. Therefore, the purpose and hypotheses of each study build upon the results and discussion of the previous study. Nesting the studies in the section as you described would eliminate this logical connection between the studies. In our eyes, this would bear the risk of reduced readability for the reader, since the purposes and hypotheses of study 2 (3) directly refer to the results and conclusions of study 1 (2, respectively). Nevertheless, this suggestion would in our eyes be of great applicability if the studies would pursue mutually exclusive goals (e.g., only one study focusing on the dimensionality, one on the convergent validity, stability, etc.) that are not consecutively logically linked among each other. Apart from that, thanks also for noting that the tables are not perfectly presented. We agree and now present the tables separately for each studies’ results in order to increase readability.

19 Overall. Your methods are generally complete with ample description of the models and modeling procedures.

1. Recommendation. While you may be applying generally accepted procedures used in cited research, consider offering a citation of standards for language adaptation of clinical measures and corresponding research designs, e.g., International Test Commission (ITC) and standards for measurement language adaptation. https://www.intestcom.org/. Thank you for this suggestion. We reviewed the ITC guidelines and can assure that the translation procedure was performed in accordance with these guidelines. We added a sentence clarifying this matter and citing the guidelines.

20 2. Recommendation. Study 1 forms used a 4-point Likert response scale, Study 2 forms expanded the response scale to a 5-point Likert. This is a nontrivial change to the forms, one which may go beyond simply obtaining higher internal consistency. At least, some discussion may be needed to explain the process underlying this finding. Furthermore, changing the response format may in fact change the measured construct. Fitting an item response model to the data to understand the response process would be very valuable, particularly for the brief form of mimic items. Note, I am not necessarily suggesting this for this set of studies as a revision, but without question, more attention is called for. Thank you for emphasizing this point. We agree that some more discussion regarding the response scale is absolutely appropriate and included it as well in the discussion section of study 1 as in the introduction and discussion sections of study 2. The internal consistency did indeed not vary particularly strongly between both studies and response scales. However, as we now explain, a five-point scale is, in our eyes, from a theoretical perspective preferable, as it avoids forcing participants to convert an actual neutral response into a high or low response, potentially increasing error variance and thereby undermining reliability and validity. We also included a reference to a recent meta-analysis, which points out the superiority of response scales using an odd number of categories (and thereby including a middle option, such as the five-point scale does) over response scales using an even number of categories. We agree that IRT analyses would be very interesting, but we are also aware of the complexity of such analyses. Therefore, we decided not to conduct them in the present study, since we deemed a thorough IRT investigation beyond the scope of the present paper, which is already long, to some extent complex and entails many analyses and findings. However, we point out to IRT analyses as a potential future avenue at the end of the general discussion.

21 3. Comment. For the convergent and discriminant validity studies, beyond a visual inspection of the measurement correlation matrices with respect to the hypothesized covariances, a more rigorous structural equation model should be fit for hypothesis testing. On the other hand, given the obtained data, fortunately for the researchers, the 'interocular traumatic test' suffices, i.e., you know it when it hits you between the eyes. Again, however, an SEM model could have been fitted to the matrices to test the hypotheses. One of countless relevant citations is, Raykov, Tenko (2011). Evaluation of convergent and discriminant validity with multitrait-multimethod correlations. British Journal of Mathematical and Statistical Psychology, 64, 38-52. Thank you for this suggestion. We have considered reporting SEM analyses, but detected some problems, while trying to implement your suggestion: The fit of a SEM that includes both the ECS and the other measures is rather poor, likely due to the lack of fit of some of the external measures’ measurement models. For example, instruments like the Interpersonal Reactivity Index or the Reading the Mind in the Eyes Test are well-known to produce inconsistent factor st

---

## [Decision Letter · Decision Letter 1]

11 Aug 2025

Dear Dr. Janelt,

Thank you for submitting your manuscript to PLOS ONE. After careful consideration, we feel that it has merit but does not fully meet PLOS ONE’s publication criteria as it currently stands. Therefore, we invite you to submit a revised version of the manuscript that addresses the points raised during the review process.

We look forward to receiving your revised manuscript.

Kind regards,

Paweł Larionow, Ph.D.

Academic Editor

PLOS ONE

Journal Requirements:

Reviewers' comments:

Reviewer's Responses to Questions

**Comments to the Author**

Reviewer #1: All comments have been addressed

Reviewer #2: All comments have been addressed

Reviewer #3: All comments have been addressed

2. Is the manuscript technically sound, and do the data support the conclusions?

Reviewer #1: Yes

Reviewer #2: Yes

Reviewer #3: (No Response)

3. Has the statistical analysis been performed appropriately and rigorously?

Reviewer #1: Yes

Reviewer #2: Yes

Reviewer #3: (No Response)

4. Have the authors made all data underlying the findings in their manuscript fully available?

Reviewer #1: Yes

Reviewer #2: Yes

Reviewer #3: (No Response)

5. Is the manuscript presented in an intelligible fashion and written in standard English?

Reviewer #1: Yes

Reviewer #2: Yes

Reviewer #3: (No Response)

Reviewer #1: The authors have addressed all the reviews and comments, and I believe that the paper, in its current form, is of sufficient quality to be published in the journal PLOS One.

Reviewer #2: Overall. Thank you for the careful and extensive response to reviewer comments. As with the initial submission, I find your research very interesting and quite likely valuable theoretically and practically. I appreciate the theoretical framework and justification for the three research studies. Overall, the research in my opinion is measurement development and validation with strengths and weaknesses. But, the missing analyses do not render your findings and interpretations indefensible. I do have lingering comments and questions regarding (a) formatting, and (b) unsubstantiated interpretations.

Title.

Comment. The Full Title is excessive, the Short Title is preferred and sufficiently accurate.

Keywords.

Comment. Because your measures focus on 'mimicry', I would consider including 'mimicry' as a keyword. Rather than validity, reliability, and factor structure, consider dropping 'factor structure', and/or replace all terms with 'psychometrics'.

Abstract.

The abstract is sufficiently comprehensive. Reference to three studies is made, but only Study 2 is mentioned specifically. The acronym 'sECS' is used, but not defined.

1. Recommendation. Do not mention any of the three studies, specifically. Consider not mentioning specific findings, e.g., "The correlation pattern . . . and longitudinal invariance." Perhaps it suffices to state that three studies using psychometric modeling and construct validation procedures concluded emotional contagion is essentially unidimensional, with secondary dimensions. Using these findings . . . a short 'mimicry' scale was developed and validated. And so on . . ..

Introduction.

Overall. The theoretical framework is clearly provided. Essential constructs and variables are identified along with the purpose(s) of the current research.

1. Comment. You start with the constructs 'emotional contagion' (and components including 'mimicry'), and 'empathy', and 'susceptibility to emotional contagion'. It is the susceptibility to emotional contagion as a personality trait that underlies the argument that measurement of 'mimicry' is meaningful.

2. Comment. You start with the constructs 'emotional contagion' (and components including 'mimicry'), and 'empathy', and 'susceptibility to emotional contagion'. It is the susceptibility to emotional contagion as a personality trait that underlies the argument that measurement of 'mimicry' is meaningful. The transition to a measure of 'mimicry' as though it is a measure of 'susceptibility of emotional contagion' is critical. This is made most clear in two places. First, you state, "Specifically, SEC can be defined as "the tendency to automatically mimic and synchronize expressions, vocalizations, postures, and movements with those of another person's and consequently, to converge emotionally". (lines 121-123) Second, the section 'Content examination of the ECS items' explains, "The ECS contains items tapping the basic process of mimicry." (line 207).

3. Suggestion. Make more clear why 'mimicry' measurement is relevant, if not identical to susceptibility of emotional contagion. Perhaps the statement above (comment 2) intends this, but the transition to 'mimicry' as the focal measure is lost when you persist with the SEC scaling discussion in 'The present studies' section. Regarding your selection of mimicry items, you state, ". . . they capture mimicry, the first state of SEC, which can be considered to reflect SEC in the most basic way." (lines 271-272). You do further explain the significance of mimicry, but it may be helpful to do this in the theoretical framework when justifying development of a 'mimicry' measure.

The Series of Three Studies Collectively.

Methods.

Overall. Your methods are generally complete with ample description of the models and modeling procedures.

1. Comment. Study 1 uses the four-point Likert scale, Studies 2 and 3 use a five-point Likert scale as described, and a sum of item responses is the total scale score for both the ECS and Mimicry measures. This modified item response scaling is critical for reasons you mention. One analysis you did not do, which would be interesting, is the study of the responses using the so called 'neutral' response. As suggested in prior comments, an item response modeling of the item responses would be interesting and you do mention this in your Discussion. You use this item response format as one possible explanation for the reliability and correlations obtained in Studies 2 and 3, relative to the relatively low values in Study 1.

2. Comment. For each study you have a section 'Internal consistency'. Consider 'Reliability' rather than 'internal consistency', even though you do use coefficient alpha. The focus of the section is on 'reliability' and the specific index. Perhaps expand the discussion to focus on item and total score reliability.

Results.

Overall. Results from analyses per study are well described and interpretation are accurate. I simply wish you had completed more item level analyses given what you think are critical implications of the item response format (4, 5 point Likert scale).

Discussion.

Overall. There are many findings to be discussed and you were quite thorough. One topic I found questionable pertained to the sample/populations you studied. Irrespective of whether you are sampling students or nurses, the actual psychometrics for ECS and mimicry should reasonably be invariant. I cannot imagine why factor structure, correlations, etc. would vary across these populations unless the measures are unreliable, which they actually are, i.e., unreliable. The low reliability is your best explanation for the variance across these populations. You did obtain acceptable convergent and discriminant validity evidence, which is somewhat unexpected given the low reliability, particularly in Study 1 with the Mimicry measure.

Writing.

Overall. The manuscript is very carefully written.

1. Major Recommendation. My primary concern pertains to the formatting within the presentation of the three studies. The bold headings and subheadings make it very difficult to appreciate the nested content organization. I suggest uses a heading and subheading formatting that clearly differentiates subheads/content.

2. Comment. Often technical terms are inconsistent with conventional nomenclature. Here are few examples.

(a). line 175 'obstructing the fit' (maybe terms such as 'attenuating', 'diminishing' the fit) are more conventional.

(b) line 220 'love should be counted as a basic emotion' (love should be considered . . .)

(c) line 234 'version a psychological meaning' (version a construct validation)

3. Comment. There are occasional spelling errors, for example,

(a) page 12 line 266 spelling error 'suceptiblity'.

Reviewer #3: (No Response)

**Do you want your identity to be public for this peer review?** For information about this choice, including consent withdrawal, please see our Privacy Policy

Reviewer #1: **Yes: ** Mohsen Khosravi

Reviewer #2: **Yes: ** Paul Yovanoff

Reviewer #3: **Yes: ** Anton Marx

---

## [Author Response · Author response to Decision Letter 2]

21 Aug 2025

(Copy of the response letter)

Editor‘s / Reviewer’s suggestions Response

Journal requirements

01 If the reviewer comments include a recommendation to cite specific previously published works, please review and evaluate these publications to determine whether they are relevant and should be cited. There is no requirement to cite these works unless the editor has indicated otherwise. Thank you for this clarification. The previous reviewer comments recommended us to provide the reader with a more comprehensive literature overview, but did not request specific citations from us. The sources we added in the first revision were deemed relevant and chosen freely by us.

02 Please review your reference list to ensure that it is complete and correct. If you have cited papers that have been retracted, please include the rationale for doing so in the manuscript text, or remove these references and replace them with relevant current references. Any changes to the reference list should be mentioned in the rebuttal letter that accompanies your revised manuscript. If you need to cite a retracted article, indicate the article’s retracted status in the References list and also include a citation and full reference for the retraction notice. We have reviewed our reference list. We did not find any retracted articles or other mistakes. We included a detailed list of any changes made to the reference list in this rebuttal letter (see below). This list of reference changes includes as well the changes made in the previous revision, as changes made in the current revision.

Reviewer #1:

The authors have addressed all the reviews and comments, and I believe that the paper, in its current form, is of sufficient quality to be published in the journal PLOS One. We thank you very much for your time and we are happy that you deem our manuscript suitable for publication.

Reviewer #2

Overall. Thank you for the careful and extensive response to reviewer comments. As with the initial submission, I find your research very interesting and quite likely valuable theoretically and practically. I appreciate the theoretical framework and justification for the three research studies. Overall, the research in my opinion is measurement development and validation with strengths and weaknesses. But, the missing analyses do not render your findings and interpretations indefensible. I do have lingering comments and questions regarding (a) formatting, and (b) unsubstantiated interpretations. Thank you again very much for reading and reviewing our manuscript and for the appreciation you expressed.

03 Title.

Comment. The Full Title is excessive, the Short Title is preferred and sufficiently accurate. We changed the title as requested.

04 Keywords.

Comment. Because your measures focus on 'mimicry', I would consider including 'mimicry' as a keyword. Rather than validity, reliability, and factor structure, consider dropping 'factor structure', and/or replace all terms with 'psychometrics'. We added “mimicry” as a keyword and generally revised the keywords in light of the shortened title.

05 Abstract.

The abstract is sufficiently comprehensive. Reference to three studies is made, but only Study 2 is mentioned specifically. The acronym 'sECS' is used, but not defined.

1. Recommendation. Do not mention any of the three studies, specifically. Consider not mentioning specific findings, e.g., "The correlation pattern . . . and longitudinal invariance." Perhaps it suffices to state that three studies using psychometric modeling and construct validation procedures concluded emotional contagion is essentially unidimensional, with secondary dimensions. Using these findings . . . a short 'mimicry' scale was developed and validated. And so on . . .. Thank you for highlighting this point. We omitted the abbreviation “sECS” from the abstract. We also avoided to merely refer to study 2 specifically. However, we maintained a description of the most important specific findings, which also include exceptional analyses beyond the typical psychometric validation procedures, which may be unexpected by and of special interest for the reader (e.g., longitudinal invariance and stability).

06 Overall. The theoretical framework is clearly provided. Essential constructs and variables are identified along with the purpose(s) of the current research.

1. Comment. You start with the constructs 'emotional contagion' (and components including 'mimicry'), and 'empathy', and 'susceptibility to emotional contagion'. It is the susceptibility to emotional contagion as a personality trait that underlies the argument that measurement of 'mimicry' is meaningful. Thank you for the appreciation you expressed. In the previous revision, we had restructured the introduction to address another reviewer’s comments recommending a more thorough introduction into the background of the core construct (susceptibility to emotional contagion). We now added a sentence at the beginning of the introduction, clearly stating that the present research indeed focuses on the susceptibility to emotional contagion, but emotional contagion is at first defined for the purpose of a thorough introduction into the topic.

07 2. The transition to a measure of 'mimicry' as though it is a measure of 'susceptibility of emotional contagion' is critical. This is made most clear in two places. First, you state, "Specifically, SEC can be defined as "the tendency to automatically mimic and synchronize expressions, vocalizations, postures, and movements with those of another person's and consequently, to converge emotionally". (lines 121-123) Second, the section 'Content examination of the ECS items' explains, "The ECS contains items tapping the basic process of mimicry." (line 207). Thank you for emphasizing this point as well. We agree that this is a crucial matter, since the relation between mimicry and SEC is complex. Both are surely two different terms, but the tendency to express mimicry can be regarded a subconstruct of SEC. Most importantly, mimicry refers to the most important and fundamental and overt part of SEC, rendering it an especially interesting potentially brief measurement approach to the total SEC construct.

In order to consequently implement the conceptualization of SEC as a basal and “primitive” trait, we decided to solely focus on items explicitly tapping this process in the short scale. We added a sentence in the section “Susceptibility to emotional contagion” as well as sentences in the section “The present studies” to clarify this matter. To illustrate our argumentation, we also inserted a short comparison to Raven’s matrices, which are conceptualized as measuring basic processes underlying the g factor of intelligence.

08 3. Suggestion. Make more clear why 'mimicry' measurement is relevant, if not identical to susceptibility of emotional contagion. Perhaps the statement above (comment 2) intends this, but the transition to 'mimicry' as the focal measure is lost when you persist with the SEC scaling discussion in 'The present studies' section. Regarding your selection of mimicry items, you state, ". . . they capture mimicry, the first state of SEC, which can be considered to reflect SEC in the most basic way." (lines 271-272). You do further explain the significance of mimicry, but it may be helpful to do this in the theoretical framework when justifying development of a 'mimicry' measure. As explained above, we added a more thorough explanation of why the mimicry items appear to be of special interest for the measurement of SEC. We do not conceptualize mimicry as an independent phenomenon of SEC – both are, as described, strongly overlapping. Surely, mimicry only refers a subprocess of SEC, yet to the most important and the most fundamental and overt subprocess and is thus of special interest to investigate.

09 The Series of Three Studies Collectively.

Methods.

Overall. Your methods are generally complete with ample description of the models and modeling procedures.

1. Comment. Study 1 uses the four-point Likert scale, Studies 2 and 3 use a five-point Likert scale as described, and a sum of item responses is the total scale score for both the ECS and Mimicry measures. This modified item response scaling is critical for reasons you mention. One analysis you did not do, which would be interesting, is the study of the responses using the so called 'neutral' response. As suggested in prior comments, an item response modeling of the item responses would be interesting and you do mention this in your Discussion. You use this item response format as one possible explanation for the reliability and correlations obtained in Studies 2 and 3, relative to the relatively low values in Study 1. Thank you for your comment. We have now conducted IRT analyses, i.e., we specified a partial credit model and item-wise plotted the category characteristic curves (CCCs). The results indicated that the CCC of the middle response category was in each case between the CCCs of the two adjoint response categories, suggesting that the neutral response is used validly by the participants (a source is cited for this interpretation). Since the paper already contains many findings, analyses, and tables and since the response format is not the focus of the paper, we decided to report the detailed findings (plots) in the supplementary material and to only briefly mention and discuss them in the main text.

10 2. Comment. For each study you have a section 'Internal consistency'. Consider 'Reliability' rather than 'internal consistency', even though you do use coefficient alpha. The focus of the section is on 'reliability' and the specific index. Perhaps expand the discussion to focus on item and total score reliability. We renamed the sections from “Internal consistency” to “Reliability”. We also added a brief discussion of the item statistics, but decided to keep this part short in light of the length of the paper and analyses.

11 Results.

Overall. Results from analyses per study are well described and interpretation are accurate. I simply wish you had completed more item level analyses given what you think are critical implications of the item response format (4, 5 point Likert scale). Thank you for the appreciation you expressed. As explained above, we now added item-level analyses (IRT), but – in light of the length of the manuscript and the extent of the already reported analyses and findings – we tried to keep this addition concise.

12 Discussion.

Overall. There are many findings to be discussed and you were quite thorough. One topic I found questionable pertained to the sample/populations you studied. Irrespective of whether you are sampling students or nurses, the actual psychometrics for ECS and mimicry should reasonably be invariant. I cannot imagine why factor structure, correlations, etc. would vary across these populations unless the measures are unreliable, which they actually are, i.e., unreliable. The low reliability is your best explanation for the variance across these populations. You did obtain acceptable convergent and discriminant validity evidence, which is somewhat unexpected given the low reliability, particularly in Study 1 with the Mimicry measure. Thanks again for highlighting this point. We added a statement in the limitations section of the general discussion, pointing out to the variance between the findings between the three studies / samples, which can be explained by the low reliability of the measures.

13 Writing.

Overall. The manuscript is very carefully written.

1. Major Recommendation. My primary concern pertains to the formatting within the presentation of the three studies. The bold headings and subheadings make it very difficult to appreciate the nested content organization. I suggest uses a heading and subheading formatting that clearly differentiates subheads/content. We revised our formatting to make the differences between adjoining levels of headings more distinct (bold 18pt, bold italics 16pt, bold 14 pt, italics 12 pt) to avoid any confusions.

14 2. Comment. Often technical terms are inconsistent with conventional nomenclature. Here are few examples.

(a). line 175 'obstructing the fit' (maybe terms such as 'attenuating', 'diminishing' the fit) are more conventional.

(b) line 220 'love should be counted as a basic emotion' (love should be considered . . .)

(c) line 234 'version a psychological meaning' (version a construct validation) We reviewed our manuscript and corrected any formulations that seemed incompatible with conventional nomenclature, including the examples you described.

15 3. Comment. There are occasional spelling errors, for example,

(a) page 12 line 266 spelling error 'suceptiblity'. We proofread our manuscript and corrected any spelling mistakes.

Additional Changes

Changes to the reference list (made in the previous revision) We added the following references:

Becker H. Some forms of sympathy: a phenomenological analysis.DP - Apr 1931. The Journal of Abnormal and Social Psychology. 1931;26(1):58-68.

Sullins ES. Emotional contagion revisited: Effects of social comparison and expressive style on mood convergence.DP - Apr 1991. Personality and Social Psychology Bulletin. 1991;17(2):166-74.

Hess U, Fischer A. Emotional mimicry: Why and when we mimic emotions. Social and personality psychology compass. 2014;8(2):45-57.

Herrando C, Constantinides E. Emotional contagion: A brief overview and future directions. Frontiers in Psychology Vol 12, 2021, ArtID 712606. 2021;12.

Harada T, Hayashi A, Sadato N, Iidaka T. Neural correlates of emotional contagion induced by happy and sad expressions. Journal of Psychophysiology. 2016;30(3):114-23.

Lin D, Zhu T, Wang Y. Emotion contagion and physiological synchrony: The more intimate relationships, the more contagion of positive emotions. Physiology & Behavior. 2024;275:114434.

Mayo O, Horesh D, Korisky A, Milstein N, Zadok E, Tomashin A, et al. I feel you: Prepandemic physiological synchrony and emotional contagion during COVID-19. Emotion. 2023;23(3):753-63.

Homan MD, Schumacher G, Bakker BN. Facing emotional politicians: Do emotional displays of politicians evoke mimicry and emotional contagion? Emotion. 2023;23(6):1702-13.

Hall JA, Schwartz R. Empathy, an important but problematic concept. The Journal of Social Psychology. 2022:1-6.

Gerdes KE, Segal EA, Lietz CA. Conceptualising and measuring empathy. British Journal of Social Work. 2010;40(7):2326-43.

Cuff BM, Brown SJ, Taylor L, Howat DJ. Empathy: A review of the concept. Emotion review. 2016;8(2):144-53.

Altmann T, Roth M. The evolution of empathy: From single components to process models. Handbook of psychology of emotions. 2013;2:171-87.

Marx AKG, Frenzel AC, Fiedler D, Reck C. Susceptibility to positive versus negative emotional contagion: First evidence on their distinction using a balanced self-report measure. PLOS ONE. 2024;19(5):e0302890.

Kevrekidis P, Skapinakis P, Damigos D, Mavreas V. Adaptation of the Emotional Contagion Scale (ECS) and gender differences within the Greek cultural context. Annals of General Psychiatry Vol 7, 2008, ArtID 14. 2008;7.

Ekman P. An argument for basic emotions. Cognition and Emotion. 1992;6(3-4):169-200.

Barrett LF. Are Emotions Natural Kinds? Perspectives on Psychological Science. 2006;1(1):28-58.

Colombetti G. From affect programs to dynamical discrete emotions. Philosophical Psychology. 2009;22(4):407-25.

Marsh HW, Hau K-T, Wen Z. In search of golden rules: Comment on hypothesis-testing approaches to setting cutoff values for fit indexes and dangers in overgeneralizing Hu and Bentler's (1999) findings. Structural equation modeling. 2004;11(3):320-41.

Burnham KP, Anderson DR. Multimodel Inference:Understanding AIC and BIC in Model Selection. Sociological Methods & Research. 2004;33(2):261-304.

Bearden WO, Sharma S, Teel JE. Sample Size Effects on Chi Square and Other Statistics Used in Evaluating Causal Models. Journal of Marketing Research. 1982;19(4):425.

Lienert GA, Raatz U. Testaufbau und Testanalyse: Beltz Verlagsgruppe; 1998.

Kusmaryono I, Wijayanti D, Maharani HR. Number of response options, reliability, validity, and potential bias in the use of the likert scale education and social scie

---

## [Editor Report · Decision Letter 2]

24 Aug 2025

Validation of the German Emotional Contagion Scale and development of a mimicry brief version

PONE-D-25-13429R2

Dear Dr. Janelt,

We’re pleased to inform you that your manuscript has been judged scientifically suitable for publication and will be formally accepted for publication once it meets all outstanding technical requirements.

Kind regards,

Paweł Larionow, Ph.D.

Academic Editor

PLOS ONE
---

## [Editor Report · Acceptance letter]

PONE-D-25-13429R2

PLOS ONE

Dear Dr. Janelt,

I'm pleased to inform you that your manuscript has been deemed suitable for publication in PLOS ONE. Congratulations! Your manuscript is now being handed over to our production team.

Kind regards,

on behalf of

Dr. Paweł Larionow

Academic Editor

PLOS ONE